# Molecular mapping in DNA-PAINT via modified Gaussian Mixture Modeling

**Rafal Kowalewski** [1,2,4], **Susanne C. M. Reinhardt** [1,2,4], **Isabelle Pachmayr** [1,3], **Shuhan Xu**[1], **Luciano A. Masullo** [1] ✉ **& Ralf Jungmann** [1,2] ✉

Super-resolution fluorescence microscopy, and specifically DNA-PAINT, provides localization precision down to ~2 nm enabling molecular-resolution imaging. To produce molecular maps of single biomolecules, their positions must be inferred from localizations stemming from single fluorescent molecules. Current clustering methods fail to exploit the full potential of the imaging method. Here, we introduce G5M, a modified Gaussian Mixture Modeling algorithm tailored to DNA-PAINT data. By incorporating prior knowledge of localization precision, spatial constraints, and DNA hybridization kinetics, G5M accurately infers true molecular positions while avoiding overfitting. In realistic simulations of dimers, G5M resolves molecules at the Rayleigh limit with a 27-fold higher recovery rate than current methods and <0.1% false positives. Applied to experimental datasets, G5M recovers full nuclear pore complex structures and detects higher-order CD20 oligomers induced by antibody treatment, outperforming conventional DNA-PAINT analysis. G5M is implemented in the open-source Picasso platform, offering an accessible solution for high-resolution, high-accuracy molecular mapping in super-resolution microscopy.

Single-molecule localization microscopy (SMLM)[1,2] overcomes the resolution limit of conventional light microscopy by switching on only a few fluorescent molecules at a time, so that their signals do not overlap[3,4]. Each molecule appears as a diffraction-limited spot[5] and by precisely determining the center of this spot, the position of the molecule can be obtained with nanometer accuracy – far beyond the diffraction limit[6]. Because individual molecules can switch on and off multiple times, they often produce several position estimates during a measurement, forming a "cloud" of localizations that reveals their spatial distribution.

Two crucial challenges arise in SMLM. The fluorophores conjugated to the molecules of interest can transition to a long-lived non-fluorescent state or photobleach, leading to a limited number of recordable localizations. Furthermore, it has been shown that the fluorophores can interact at distances closer than 10 nm, resulting in biased signal detection[7,8].

In DNA-PAINT (DNA point accumulation for imaging in nanoscale topography)[9,10], single-molecule blinking is achieved by transient and repetitive binding of short, dye-labeled DNA oligonucleotides (imager strands) to a target-bound complement (docking strand). Localizations are recorded during the bound state, while the rapid diffusion of unbound strands results in background signal (Fig. 1a). When combined with optical sectioning, DNA-PAINT yields a typical signal-to-background ratio larger than ten[11].

Since dye-labeled imager strands are continuously replenished, photobleaching practically does not limit the image acquisition time leading to a high number of localizations obtained from every imaged biomolecule. Additionally, the absence of fluorophore-fluorophore interactions ensures that neighboring molecules (<10 nm) can be detected accurately. As a result, each molecule gives rise to a normally distributed localization cloud centered around its true position (Fig. 1b, Supplementary Fig. 1 and Supplementary Information

[1]Max Planck Institute of Biochemistry, Planegg, Germany. [2]Faculty of Physics and Center for Nanoscience, Ludwig Maximilian University, Munich, Germany. [3]Department of Chemistry and Biochemistry, Ludwig Maximilian University, Munich, Germany. [4]These authors contributed equally: Rafal Kowalewski, Susanne C. M. Reinhardt. ✉e-mail: masullo@biochem.mpg.de; jungmann@biochem.mpg.de

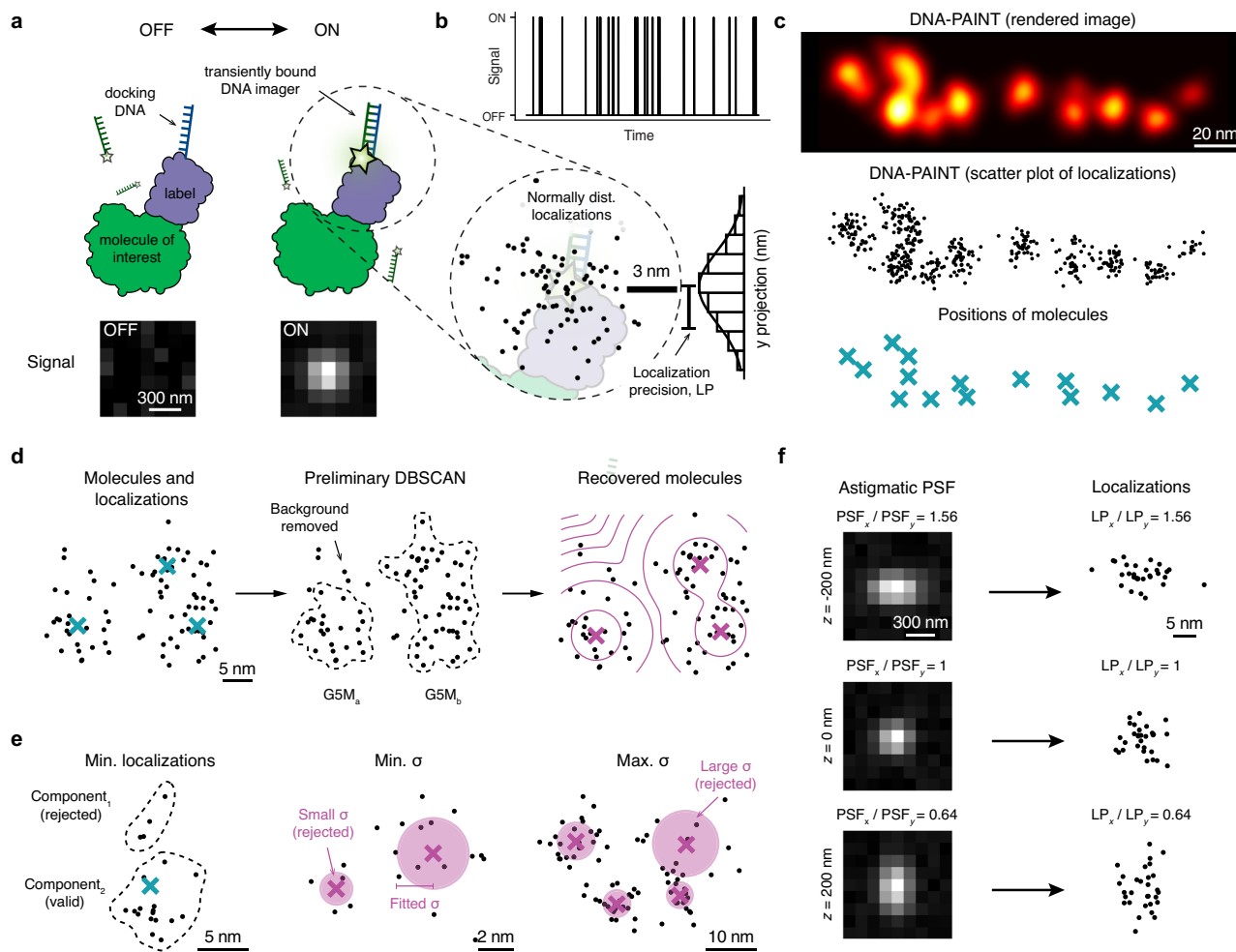

**Fig. 1 | Gaussian Mixture Modeling with Modifications for Molecular Mapping (G5M) in DNA-PAINT. a** Single-molecule blinking in DNA-PAINT is achieved via transient and repetitive binding of dye-labeled DNA imager strands to target-bound complements. Diffusing single-stranded DNA moves fast, emitting a low-amplitude signal. When bound, the fluorophore attached to DNA remains in place across several camera frames, emitting a strong signal. From each such event, a localization is found estimating the position of the molecule. The resulting normally distributed localization cloud (black points) is centered around the imaged molecule, with its standard deviation equal to localization precision (LP). **b** Exemplary trace from one molecule yielding a number of localizations across the image acquisition. **c** Common rendering of DNA-PAINT data relying on blurring each localization by its associated LP. Based on the underlying localizations (scatter plot), molecule positions are found (cyan crosses). **d** Our workflow involves two steps: localizations are

clustered with DBSCAN to split data into smaller subsets. Next, G5M is applied to each DBSCAN cluster separately such that the mean positions (μ) of the resulting Gaussian components are the fitted molecule positions (magenta crosses), leading to a molecular map. The contour plot depicts the probability density function of the Gaussian Mixture. **e** User parameters in G5M. Min. localizations ensures that the apparent sites with too few localizations are rejected. Min. σ ensures that G5M does not hallucinate molecules with unrealistically narrow spread of localizations. Max. σ rejects components that are fitted to background localizations. **f** G5M allows for 3D modeling of localization clouds resulting from an astigmatic Point Spread Function (PSF) due to a cylindrical lens in the optical path. The ratio between PSF's width and height (PSF$_x$, PSF$_y$, respectively) is approximately proportional to the associated localization cloud's width and height (LP$_x$, LP$_y$, respectively). All localizations shown are simulated.

section 1) with the spread of this distribution determined by the localization precision.

The spatial resolution achievable in DNA-PAINT and its related method RESI (Resolution Enhancement by Sequential Imaging) is exemplified by datasets in which individual biomolecules are clearly resolved in intact cells[12–15]. This represents a shift towards a new generation of SMLM datasets focused on generating "molecular maps"— precise coordinates of individual molecules—rather than simply plotting localization events (Fig. 1c). Such data can lead to novel biological insights by quantifying oligomerization states of proteins in situ[14], discovering patterns for super-selective targeting[16,17] and screening for potential patterned therapeutics[18–20].

Quantification of resolution (△x) in SMLM is a subject of debate[21], however, a commonly used metric is the full-width-at-half-maximum (FWHM) criterion, which links localization precision (LP) to the

potential ability to resolve two adjacent point-like objects (e.g. labeled biomolecules) according to $\triangle x_{FWHM} \approx 2.355\,LP$. We argue that this is an overly optimistic metric as targets at this distance cannot be confidently resolved in realistic experimental situations and that a better metric is the Rayleigh criterion[22], more commonly used in general optics and microscopy. By translating the Rayleigh criterion to two Gaussian-distributed clouds of localizations (Supplementary Fig. 2 and Supplementary Information section 2), we obtain $\triangle x_{Rayleigh} \approx 2.9\,LP$ which is a more realistic estimation for the minimum localization precision needed to distinguish two point-like objects.

Importantly, the effective capability of DNA-PAINT to produce accurate molecular maps is determined not only by the optical setup and sample preparation but also by the computational tools used to process localization data into accurate biomolecule (e.g. protein) positions. Notably, false positive (FP) errors, i.e., hallucination of non-

existent molecules, must be avoided as they can lead to mis-interpretations—including artificial oligomerization states or spurious nanoscale patterns—that undermine the biological validity of the data.

The state-of-the-art method[13,23], here referred to as "Gradient Ascent" (GA), identifies high-density regions by selecting localizations with the most neighbors within a user-specified radius and estimates molecular positions based on their spatial context. While this approach is computationally efficient, has a low FP rate and requires only a few intuitive user-defined parameters, it has a critical limitation: it fails to resolve molecules at the Rayleigh limit. These missed detections constitute false negative (FN) errors, as resolvable molecules are not recognized. This limitation can only be mitigated by employing the sparse sequential readout strategy of RESI[13], at the cost of more complex sample preparation and increased acquisition time.

Notably, dedicated algorithms for efficient molecular mapping have not been investigated in detail, possibly due to the fact that single-protein-resolution (sub-10-nm) data has only recently emerged[12–15]. Popular clustering algorithms like DBSCAN (Density-based spatial clustering of applications with noise[24]) are of general purpose and do not fully harvest the available prior information about the measurement process inherent to the localization of the fluorescent molecules, making them suboptimal for the specific task.

A probabilistic modeling approach such as Gaussian Mixture Modeling (GMM) algorithm has the potential to optimize the mapping process by accounting for the prior information available (e.g. shape of the Point Spread Function, Gaussian distribution of the localizations, DNA binding kinetics). While GMM has been applied to SMLM data in specific contexts[25,26] a general approach for efficiently mapping localizations into molecular positions via GMM is still missing. Here, we present an easy-to-use algorithm—Gaussian Mixture Modeling with Modifications for Molecular Mapping (G5M)—designed to accurately reconstruct molecular maps from DNA-PAINT datasets. G5M extends conventional GMM by incorporating modifications tailored to the characteristics of DNA-PAINT data. In a benchmark based on realistic simulations of dimers (20 localizations per molecule), G5M reliably resolves protein positions of neighboring molecules at the Rayleigh limit, with an at least 27-fold improvement in the recovery rate compared to the current methods. G5M further achieves a > 70% improvement in effective resolution. Crucially, G5M maintains a low FP error rate (< 0.1%), preventing spurious assignment of non-existent molecules.

We demonstrate the performance of G5M on simulated data, DNA origami structures[9,27,28], nuclear pore complexes[29], and oligomers of CD20 in intact cells[30]. We also provide important guidelines for experimental design and robust applicability of G5M. Importantly, to facilitate quick adoption, G5M is fully integrated into the Picasso software package for DNA-PAINT data analysis[10], with a detailed user guide provided in the Supplementary Information.

## Results

### Concept and algorithm

Before applying G5M, a preprocessing step is required to divide the data into smaller subsets of a few molecules, in order to enhance the accuracy and computational efficiency of the subsequent molecular assignment (Fig. 1d). This partitioning is performed using the DBSCAN algorithm[24], which additionally filters out isolated localizations likely stemming from non-specific signal.

Next, each DBSCAN-clustered subset is analyzed using a Gaussian Mixture Model (GMM) fitted with several modifications harvesting a priori knowledge of the DNA-PAINT measurement, i.e., the expected shape of the distribution of the localizations in space and time. The GMM is composed of a set of Gaussian components, each of which corresponds to an individual molecule; their means ($\mu$) define the molecules' positions, while their standard deviations ($\sigma$) reflect the localization precision. The number of Gaussian components is

determined using the Bayesian Information Criterion (BIC)[31], which weighs the goodness of the fit against the complexity of the model, i.e., the number of components.

To minimize overfitting, three user-defined parameters are introduced: a minimum number of localizations ($N_{locs,min}$) required for a valid molecular assignment; a minimum and maximum $\sigma$ to exclude unrealistically narrow and wide Gaussians, respectively (Fig. 1e, Supplementary Fig. 3). Importantly, localization precision can be determined accurately[32], providing strong a priori knowledge which can be harvested to determine the values of minimum and maximum $\sigma$. Furthermore, if any two components violate the Sparrow limit[33], i.e. no minimum in probability density function between them is found, such a model is rejected (Supplementary Fig. 4a, b). Lastly, the components can be filtered out based on the log-likelihood of the model given the data, as described in Supplementary Information section 3.3.2 and Supplementary Fig. 4c, d.

In 2D applications, G5M enforces spherical covariance matrices, assuming equal variance in x and y and no spatial correlation—an assumption consistent with the isotropic nature of 2D localization clouds. For 3D imaging using an astigmatic point spread function (PSF)[34], however, G5M models anisotropic localization clouds by preserving the width-to-height ratio that encodes the axial position of the molecule. Since localization precision scales with PSF size[32], this modification allows for accurate estimation of molecular positions in 3D (Supplementary Information section 3.3.3 and Fig. 1f).

G5M is adapted from the Gaussian Mixture Model implementation provided by Scikit-learn, a widely adopted Python library for machine learning and statistical analysis[35] with modifications as described above. To enhance performance, key functions were further optimized using Numba[36], and the entire procedure was parallelized via Python's multiprocessing framework. These enhancements reduce computational time (Supplementary Table 1), enabling fast and scalable processing of large DNA-PAINT datasets. A comprehensive description of the algorithmic workflow, along with pseudocode and recommended user parameters, is provided in the Supplementary Information.

### G5M outperforms other algorithms in realistic simulations and in DNA origami

To benchmark the performance of G5M, we compared it against several publicly available approaches that can be used for molecular mapping (DBSCAN[12,37], ToMATo[37,38], and GA[13,23]) using realistic simulated DNA-PAINT datasets. We first generated 2D simulations of two molecules, mimicking binding kinetics, photon count fluctuations, camera shot noise and the localization process from single-molecule images (see Methods). Therefore, pairs of molecules with varying numbers of localizations per molecule and localization precisions were obtained (Fig. 2a-c, Supplementary Fig. 5). Here, we define the spacing between the simulated molecules in terms of the median simulated localization precision ( ~ 2.7 nm). Each simulation was repeated 10,000 times at distances from 1.0 LP to 6.0 LP at steps of 0.1 LP.

The different algorithms were applied to the simulated datasets and the numbers of recovered molecules were recorded. The simulations in which 2 molecules were found were considered as true positive (TP) (Fig. 2d), if more than 2 molecules were found, they were considered as false positive (FP). To quantify the resolution, sigmoid curves were fitted to the TP curves (Supplementary Table 2). The user parameters used throughout this work were min. localizations of 10 for all algorithms; min. $\sigma$ of 0.8 LP and max. $\sigma$ of 1.5 LP for G5M and clustering radius of 2 LP for GA (see Methods).

GA achieved a maximum 77% true positive (TP) rate and 99% of the maximum TP rate was achieved at the distance of 5.3 LP, far above the Rayleigh limit (Supplementary Fig. 2). Given that on average 20 localizations per molecule were simulated, it is expected that a certain fraction of simulated molecules produces as little as 10 localizations (Supplementary Fig. 6), not all of which are found within the user-

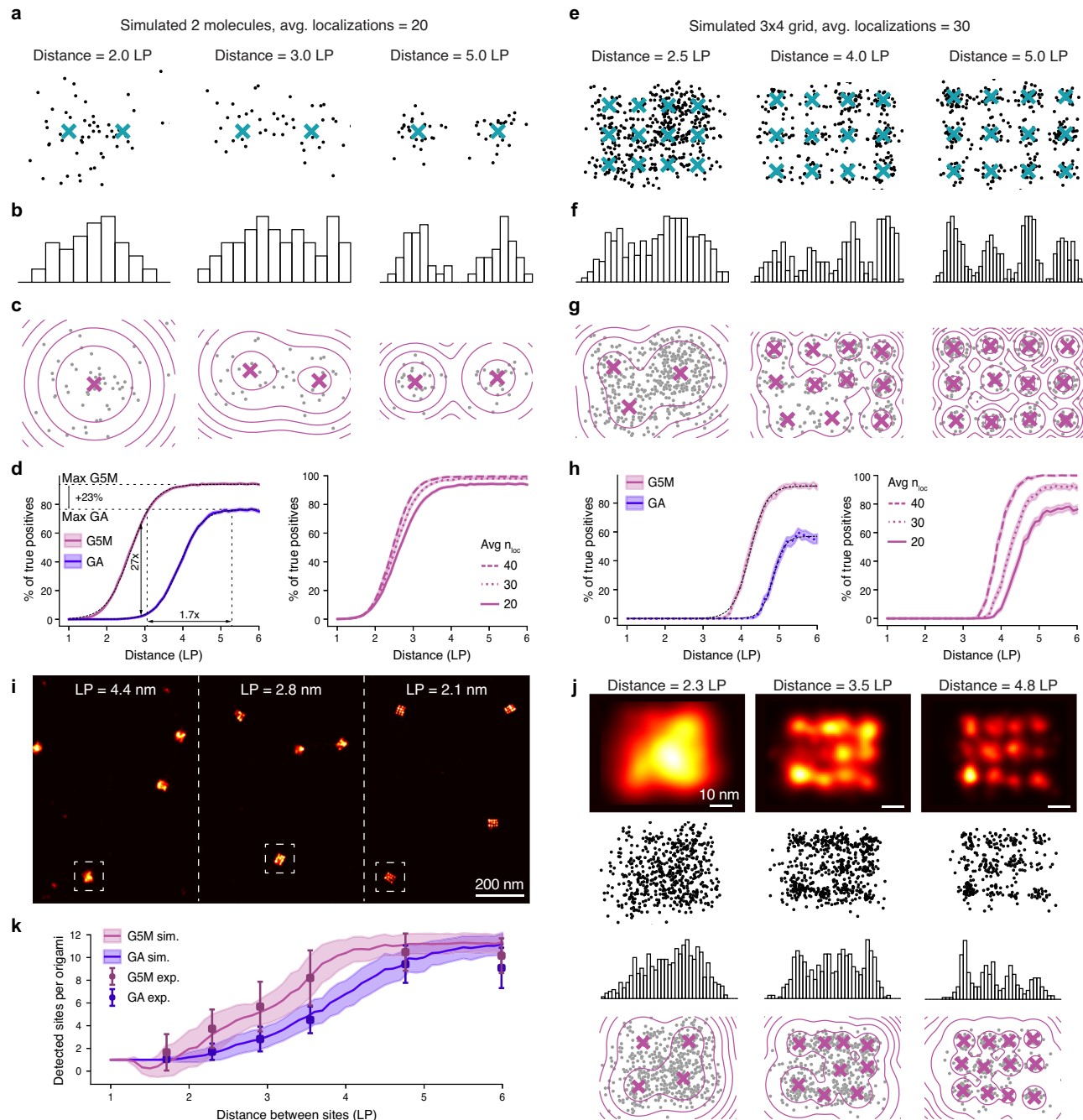

**Fig. 2 | G5M accurately recovers molecular maps in DNA-PAINT. a–c** 2 molecules were simulated 10,000 times across different distances, expressed in terms of median localization precision (LP), ranging from 1.0 LP to 6.0 LP, with average 20 localizations per molecule. Three cases are shown: 2.0 LP, 3.0 LP and 5.0 LP with the simulated molecules and localizations (**a**), x-axis projection of localizations (**b**) and G5M-fitted molecules (**c**). **d** True positive (TP) rate for G5M (magenta) and Gradient Ascent (GA, blue) are plotted. The thick lines display the median TP rate and the envelope depicts the 95% CI, bootstrapped 1000 times. The true positive rate for G5M when changing the average number of localizations is shown on the right. **e–h** Similar to (**a–d**) but with 12 molecules simulated in a 3 × 4 grid and average 30

localizations per molecule. **i** DNA origami sample data, similar to the 3 × 4 grid from (**e–h**). The grids where sites are spaced by 10 nm were imaged at different laser powers, leading to different localization precisions, each measured $N = 1$ times. **j** Example zoom-ins to 3 origamis with underlying localizations and G5M results. **k** Number of detected sites per origami at different distances between the sites. Simulation (sim.) results are shown by the continuous lines with their corresponding envelopes, while the experimental (exp.) data results are represented by the scatter plots with their corresponding error bars. Mean values ± 1 SD are displayed. Results are shown for $N = (247, 270, 343, 254, 302, 360)$ origamis per laser power, respectively. Source data are provided as Source Data file.

defined clustering radius of 2 LP. This leads to the 23% false negative (FN) rate. FP rate was very low (< 0.1%) for all simulations.

On the other hand, G5M achieved a maximum 94% TP rate. The 6% FN rate is due to the preliminary DBSCAN step that occasionally misses some of the localizations, especially when the number of localizations

produced by a molecule is low (Supplementary Fig. 6). 99% of the maximum TP rate was achieved at the distance of 4.1 LP, thus recovering 23% more molecules compared to GA at distances >5.0 LP. G5M achieved GA's maximum TP rate at 3.1 LP. Therefore, G5M improves the resolution in DNA-PAINT over the state-of-the-art by 73%.

Moreover, at the Rayleigh limit (2.9 LP), G5M accurately recovered 2 molecules in 67% of simulations, while GA only in 2.5%. Thus, at the resolution limit for two adjacent molecules, G5M improves the recovery rate 27-fold. Moreover, G5M maintains a negligible FP rate (< 0.1%). Notably, the recovery rate improves up to 99% when more localizations per molecule are detected (Fig. 2d, Supplementary Fig. 5, Supplementary Table 2), highlighting the importance of obtaining a higher number of localizations (average 30 or 40 per molecule), which can be achieved in DNA-PAINT by increasing the acquisition time and imager concentration[10]. Furthermore, we evaluated the precision of the recovered molecule positions (Supplementary Fig. 7a–c). Finally, we note that our results are robust against variations from the default user parameters (Supplementary Information section 3.4 and Supplementary Fig. 8 and 9).

Other tested algorithms for molecular mapping performed worse than GA and G5M (Supplementary Fig. 10). DBSCAN achieved high TP rate (82%) only at the distance of 6.0 LP, while in the case of ToMATo, FP rate was significantly higher (up to 1.5%). Importantly, the two methods require careful fine-tuning of the user parameters, thus the recovery rates in real-world data may be strongly affected by their suboptimal values.

We then evaluated the algorithms on a denser target geometry: a 3 × 4 grid of 12 molecules (Fig. 2e–g). Here, we simulated on average 30 localizations per molecule since such oligomeric structures are expected to be more difficult to resolve[39]. G5M achieved maximum TP rate of 92%, while GA achieved 57% (Fig. 2h, Supplementary Table 3). The FP rate of G5M increased to up to 0.2%, while GA showed <0.1% FP rate. GA achieved 99% of its maximum TP rate at the distance of 5.5 LP, which for G5M was 5.2 LP. G5M achieved GA's max. TP rate at the distance of 4.3 LP, indicating ~1.3-fold improvement. Other algorithms achieved much lower TP rates or showed high FP rates (Supplementary Fig. 11).

To validate these findings experimentally, we designed DNA origami structures containing twelve 10 nm-spaced docking sites arranged in a 3 × 4 grid, mimicking the simulated geometry. By varying the excitation laser power during imaging of the same sample, we generated datasets with different localization precisions (Fig. 2i–j). Accounting for an expected ~6% of missing sites due to incomplete DNA incorporation[14,40], each origami was expected to contain on average 11.3 docking sites. G5M outperformed GA across the probed

range between ~2 LP and ~6 LP (Fig. 2k). These results are further supported by simulations (solid line) that include a realistic description of the DNA origami structures (see Methods).

In 3D, we simulated two molecules separated in the axial direction, analogous to Fig. 2a–d. Two scenarios were simulated assuming astigmatic imaging: one centered around $z = 0$ nm, where width and height of a single-molecule image are roughly equal and one at $z = 260$ nm where the width is larger than the height. Importantly, the size of the localization cloud in $z$ changed between the two scenarios, as predicted by our formula for axial localization precision (Supplementary Information section 4, Methods and Supplementary Fig. 12a).

We compared G5M against GA and found significant improvement for G5M (Supplementary Fig. 12b, c). At $z = 0$ nm, G5M achieved the maximum TP rate of 88%, while GA only 62%. At $z = 260$ nm, G5M achieved the maximum TP rate of 95%, while GA did not plateau within the inspected dimer distance range of (1 LP–6 LP). Both algorithms maintained very low FP rates (< 0.1%). GA's poor performance can be attributed to its lack of flexibility in handling the variety of sizes of localization clouds due to its rigid clustering radius. On the other hand, G5M utilizes the local localization precision to determine the shape of the fitted Gaussian components.

## G5M outperforms current methods in 3D biological context

To demonstrate the applicability of G5M in a biologically relevant 3D context, we applied it to nuclear pore complexes (NPCs), focusing on Nup96 – a scaffold protein with a well-characterized spatial arrangement[41–43]. NPCs are large, symmetric protein assemblies embedded in the nuclear envelope that facilitate nucleocytoplasmic transport[44]. Their regular structure and known geometry make them ideal benchmark targets for super-resolution microscopy[29].

Based on the CryoET data[45], Nup96 is found on two concentric rings in the NPC – cytoplasmic (CR) and nuclear (NR) – positioned on either side of the nuclear envelope and spaced approximately 50 nm apart. Each ring contains 16 Nup96 proteins arranged in eight symmetrically spaced pairs, forming a circular structure with an overall diameter of ~105 nm (Fig. 3a, b). To visualize this architecture, we performed DNA-PAINT on U2OS cells expressing GFP-tagged Nup96, labeled with anti-GFP nanobodies. We obtained a median localization precision of ~2.7 nm in the CR and ~3.3 nm in the NR. At this resolution, the expected ~11 nm separation between adjacent Nup96 pairs

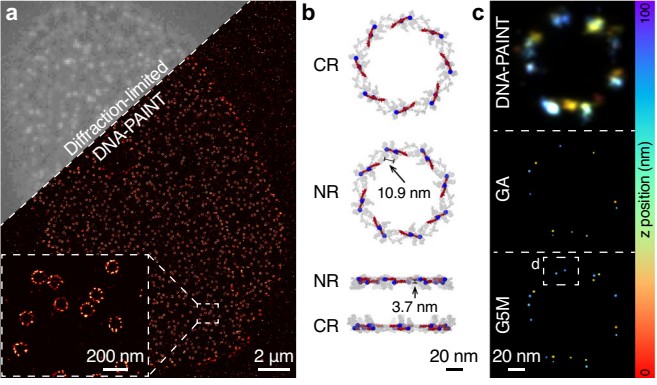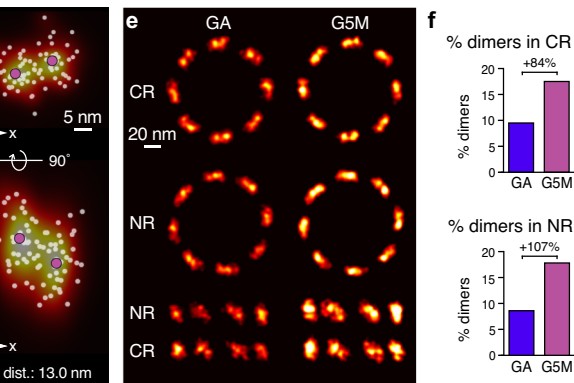

**Fig. 3 | G5M successfully recovers the structure of Nup96 in NPCs. a** DNA-PAINT measurement of Nup96-GFP in N = 1 U2OS cell, with comparison to the widefield, diffraction-limited image. **b** Cryo-ET structure of NPC, Nup96, and its C-termini (gray, red, and blue, respectively), showing an 8-fold symmetry of Nup96 across the two rings: nuclear ring (NR) and cytoplasmic ring (CR). The distance between two adjacent Nup96 copies is 10.9 nm. Adapted from PDB 7PEQ. **c** Localizations from one NPC with z position color-coded and molecular maps obtained with Gradient Ascent (GA) and G5M. G5M recovered 18 molecules, while GA only 11. In particular, the highlighted dimer was missed by GA. Scale bar = 20 nm. **d** Zoom-in to the highlighted pair of adjacent Nup96 molecules (magenta dots) from (**c**) on top of the

associated localizations: both standard rendering (hot colormap) and scatter plot of localizations (white dots). The 3D measured distance between the two molecules is 13.0 nm. **e** LocMoFit alignment of GA- and G5M-recovered molecules from NPCs from a single nucleus (N = 1007 and 1059 NPCs for GA and G5M, respectively). G5M recovers the structure from (**b**). The side views display 4 of the 8 segments for better visibility. **f** Dimer counts of recovered molecules per each of the 8 segments in individual rings within single NPCs, based on the alignment in (**e**). In both rings, G5M shows a roughly 2-fold increase in dimer detection. All DNA-PAINT results come from N = 1 cell. Source data are provided as Source Data file.

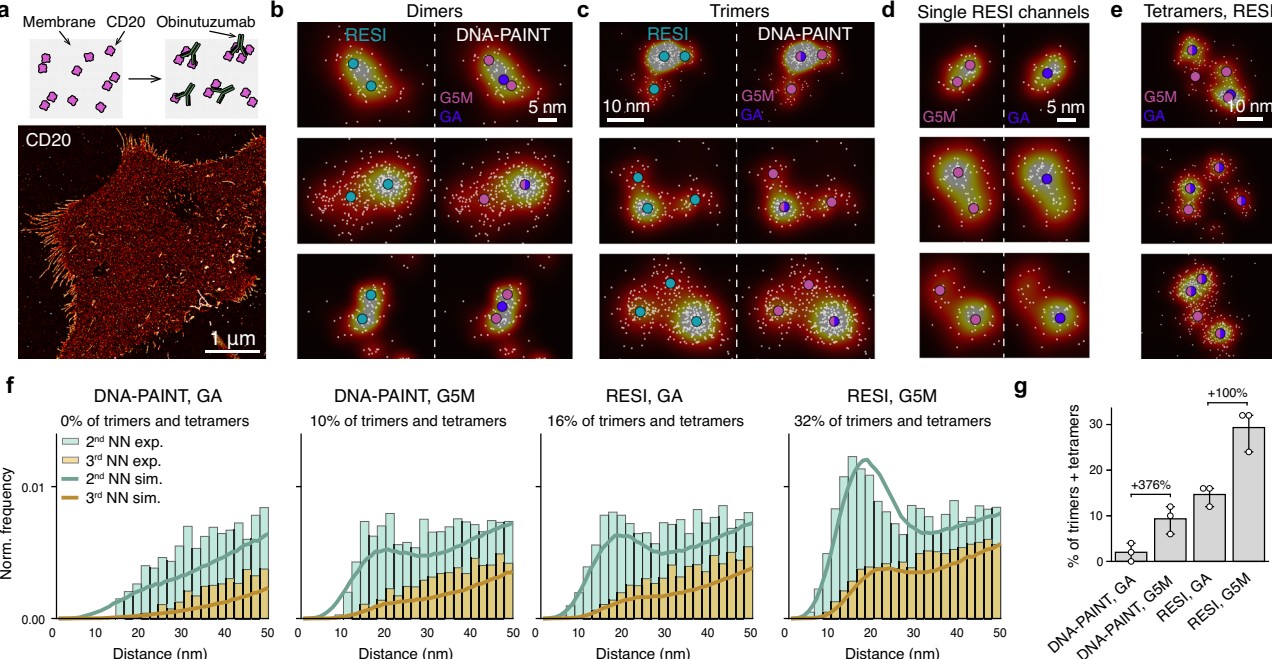

**Fig. 4 | G5M recovers more oligomers in Obinutuzumab-treated CD20 compared to the state-of-the-art. a** Schematic plot showing the oligomerization (up to tetramers) of CD20 upon Obinutuzumab binding and example imaged cell with CD20 signal. **b** Example fields of view (FOVs) where RESI showed a dimer (teal dots) that was correctly assigned in the DNA-PAINT dataset using G5M (magenta dots) but not with Gradient Ascent, (GA, blue dots). Underlying localizations are rendered (hot colormap) together with their scatter plot (white dots). **c** Same as (**b**) but showing trimers. **d** Example FOVs where within individual RESI channels only one molecule was detected with GA but 2 with G5M, highlighting the potential of G5M to improve RESI. **e** Example FOVs showing how the improvement showcased in (**d**) allows for better tetramer detection using G5M compared to GA. **f** Simulated (sim.) and experimental (exp.) 2nd and 3rd nearest neighbor (NN) distances show that DNA-PAINT with GA detects less oligomers than DNA-PAINT with G5M or RESI. SPINNA simulations quantify the improvement in the proportions of detected trimers and tetramers. **g** Summary results for $N = 3$ imaged cells, showing that DNA-PAINT with G5M detected 376% more higher-order oligomers, however, RESI offers extra improvement in resolution. Bars show the mean value and the error bars −95% CI. Source data are provided as Source Data file.

corresponds to roughly 4.1 LP and 3.3 LP in the CR and NR, respectively – challenging for standard molecular mapping tools, however, feasible for G5M (Fig. 3c, d).

To assess accuracy on a larger scale, we used LocMoFit[46] to align the GA- and G5M-derived molecular positions from ~1000 NPCs within a single cell, without imposing any structural model a priori. The resulting reconstruction (Fig. 3e) was successful for GA in the CR, however, it failed in the NR. On the other hand, the resulting reconstruction for G5M closely matched previously reported NPC geometry in both rings[13,45], confirming that G5M accurately recovers molecular positions in complex 3D biological environments.

Additionally, to further demonstrate the improvement in resolution, we split the aligned molecules in each ring into 8 groups, such that each Nup96 dimer was separated (Supplementary Fig. 13). Therefore, we could establish how many molecules in each segment were recovered on a single NPC basis (Supplementary Table 4). Labeling efficiency (LE) affects how many molecules were imaged per segment, thus in only a fraction of cases (f) a dimer is expected to be observed ($f = LE^2$). For G5M, the fraction of dimers detected per segment was 17% and 18% in the CR and the NR, respectively. For GA: 10% and 9%, thus G5M recovered almost twice as many dimers (Fig. 3f). Using the approach analogous to Thevathasan et al.[29], we used these values to see if we could retrieve the previously reported labeling efficiency of the GFP nanobody. For GA, the number obtained was ~33% while for G5M, ~42%, close to the previously reported value[47] of (47 ± 2) %, showcasing the enhanced accuracy of G5M. Moreover, among the segments with labeled molecules, 98% contained one or two molecules (Supplementary Table 4), in agreement with previously reported cryo-EM data showing Nup96 pairs.

## G5M enhances the detection of oligomerization of membrane receptors

Next, we applied G5M to the study of the organization of CD20 membrane receptors, common targets for therapeutic antibody treatment of B-cell-derived blood cancers and autoimmune diseases[48]. We focused on Obinutuzumab, a Type II monoclonal antibody that is known to induce limited CD20 oligomerization, reaching up to tetramers[49,50] (Fig. 4a). In our recent study[30] we imaged CD20 using RESI[13] (Fig. 4a), where a single target species is separated into multiple, sparser subsets labeled with orthogonal DNA sequences. By imaging the subsets sequentially using DNA-PAINT, sufficiently spaced and isolated groups of localizations are measured. Determining the center of each group of localizations yields a resolution enhancement. This grouping of localizations has so far been performed using GA.

By correlating DNA-PAINT and RESI images of the same dataset, we could directly compare GA and G5M in DNA-PAINT (median localization precision ~3 nm) using RESI (median precision ~0.6 nm) as the ground truth. As expected, we observed cases in which dimers and trimers were detected by RESI and DNA-PAINT using G5M but omitted in DNA-PAINT using GA (Fig. 4b, c).

Moreover, G5M can further improve the quality of RESI by enhancing the resolution within individual channels. For example, we observed several cases where only G5M and not GA detected dimers (Fig. 4d). Downstream, this allowed for detecting CD20 oligomers with higher sensitivity using RESI with G5M, as shown in the examples in Fig. 4e. Additionally, we conducted several tests showing that no significant false positive errors were present. Firstly, we confirmed that the distances measured between molecules originating from the same RESI round match the distances between molecules from different

RESI rounds (Supplementary Fig. 14). Moreover, if false positive errors are present, the number of binding events per molecule is expected to be higher for molecules in isolated than in dense environments. We observed roughly equal counts, further indicating the absence of the errors (Supplementary Fig. 15). While the first test is only applicable to RESI datasets, the second test is automatically performed when running G5M in Picasso. Lastly, we provide a gallery of example dimers recovered in each RESI round for visual inspection (Supplementary Fig. 16).

We quantified the percentages of trimers and tetramers (referred to as oligomers) observed with each technique using SPINNA[14] (Fig. 4f). DNA-PAINT with GA retrieved no oligomerization while DNA-PAINT with G5M retrieved 10%. As expected, RESI further enhances resolution and by combining it with GA, 16% oligomerization was observed. Furthermore, RESI with G5M led to 32% oligomerization, suggesting that G5M enhances also the performance of RESI. Figure 4g summarizes the results for $N = 3$ cells. In DNA-PAINT, G5M increases the oligomer detection rate over GA nearly 4-fold (376%), while in RESI by 100%. Moreover, RESI with G5M revealed that the inverted conformation of Obinutuzumab-based CD20-CD3-T cell engagers[51] induces CD20 oligomerization more effectively than Obinutuzumab or the classical T cell engager conformation (Supplementary Fig. 17), in agreement with our recent findings[30].

## Discussion

We have developed G5M, a modified Gaussian Mixture Modeling (GMM) algorithm tailored for molecular mapping in DNA-PAINT. G5M addresses key limitations of existing approaches by modeling the shape of localization clouds and leveraging probabilistic inference to resolve closely spaced molecules with high accuracy. Through a combination of expectation-maximization, model selection via the Bayesian Information Criterion (BIC) and usage of a priori information, G5M achieves both high sensitivity and specificity in molecular assignment, outperforming widely used clustering-based methods that fail to separate overlapping signals.

A major strength of G5M lies in its ability to recover molecules spaced at distances significantly lower than other algorithms, unlocking previously inaccessible spatial information from standard DNA-PAINT datasets. In silico, G5M maintains a near-zero false positive (FP) error rate, and in experimental data it accurately recovers the expected number of docking sites in both DNA origami structures and cellular environments. In NPCs, G5M identifies one or two Nup96 molecules in 98.4% of cases, consistent with previously reported structural data[45]. Together, these results show that BIC in combination with the constraints on $\sigma$, $N_{locs,min}$, log-likelihood, minimum separation and 3D localization cloud shape correctly identify the number of components for the Gaussian Mixture Model (Supplementary Fig. 3). In particular, they minimize the risk of artificially inflating oligomeric states or introducing spurious patterns. This renders G5M suitable for quantitative applications, including stoichiometry analysis and spatial pattern recognition.

We demonstrated the superior performance of G5M through simulations involving sparse (2-molecule) and dense (12-molecule) scenarios and compared it to current methods, in particular, Gradient Ascent (GA). G5M resolved the molecules at closest spacing while also showing the highest true positive (TP) rate at large separations. Our simulations provide a guide to the user demonstrating the expected performance of G5M dependent on the complexity of the data (dimer vs. grid), localization precision and the number of localizations per molecule. To ensure the applicability of G5M, data points from single molecules must follow a normal distribution. Accordingly, drift-corrected localizations are suitable for G5M analysis, whereas linked localizations violate this requirement (Supplementary Information Section 1).

The superior performance of G5M in molecular mapping observed in silico and in vitro translated directly to improved biological imaging. Applying G5M to nuclear pore complexes resulted in the accurate recovery of the known NPC architecture. With an average localization precision of 3.3 nm, G5M reliably resolved the Nup96 pairs, also in the nuclear ring. G5M detected a fraction of the Nup96 dimers consistent with our expectations based on the labeling efficiency (LE) value of ~47%. On the other hand, GA recovered only half of the dimers, showcasing the improved resolution in realistic biological settings. Unlike earlier efforts that relied on optical or biochemical optimization[13,52], our improvements stem purely from software advances which can be readily applied even to already existing datasets retrospectively.

Lastly, we showed that G5M outperforms GA in DNA-PAINT data of Obinutuzumab-treated CD20, resolving more oligomers in the data. Notably, G5M can also further boost existing resolution-enhancement strategies such as RESI[13]. By improving molecular assignment even in densely labeled conditions, G5M enhances the overall resolution and information content of these advanced techniques.

G5M is robust to parameter variation (default parameters are sufficient in most common scenarios) and it is fully integrated into the Picasso software package[10], facilitating seamless adoption by the super-resolution community. Downstream analyses—including nearest-neighbor statistics[12] and SPINNA[14]—can directly benefit from the high-quality molecular maps generated by G5M. We anticipate that G5M will become a standard tool for single-protein-resolution DNA-PAINT data analysis and will facilitate a deeper understanding of nanoscale molecular organization in biological systems.

## Methods

### Materials

Unmodified DNA oligonucleotides were purchased from Integrated DNA Technologies. DNA oligonucleotides modified with C3-azide and Cy3B were ordered from Metabion and MWG Eurofins. M13mp18 (p7249) scaffold was purchased from Tilibit. Magnesium chloride (1 M; AM9530G), ultrapure water (10977-035), Tris (1 M, pH 8; AM9855G), EDTA (0.5 M, pH 8.0; AM9260G) and 10× PBS (70011051) were purchased from Thermo Fisher Scientific. Sodium chloride (5 M; S5150-1L) and BSA (A4503-10G) were ordered from Sigma-Aldrich. Triton X-100 (6683.1) was purchased from Carl Roth. Sodium hydroxide (31627.290) was purchased from VWR. Paraformaldehyde (15710) and glutaraldehyde (16220) were obtained from Electron Microscopy Sciences. Tween-20 (P9416-50ML), glycerol (65516–500 ml), methanol (32213–2.5 L), protocatechuate 3,4-dioxygenase pseudomonas (PCD; P8279), 3,4-dihydroxybenzoic acid (PCA; 37580-25G-F) and (±)-6-hydroxy-2,5,7,8-tetra-methylchromane-2-carboxylic acid (Trolox; 238813-5 G) were ordered from Sigma-Aldrich. Neutravidin (31000) was purchased from Thermo Fisher Scientific. Biotin-labeled BSA (A8549) and Sodium azide (769320) were obtained from Sigma-Aldrich. Double-sided tape (665D) was ordered from Scotch. FBS (cat. A5669701, Gibco), 0.05% trypsin–EDTA (Gibco, no. 253000-054), Salmon Sperm DNA (15632011), Lipofectamine LTX (A12621) were purchased from Thermo Fisher Scientific. Ninety-nanometer gold nanoparticles (G-90-100) were ordered from Cytodiagnostics.

### Buffers

The following buffers were used for sample preparation and imaging:
- 1x DNA origami folding buffer: 10 mM Tris, 1 mM EDTA, 12.5 mM $MgCl_2$; pH 8
- FoB5 buffer: 5 mM Tris, 1 mM EDTA, 5 mM NaCl, 5 mM $MgCl_2$; pH 8
- Buffer A: 10 mM Tris-HCl pH 8, 100 mM NaCl, and 0.05% Tween-20; pH 8
- Buffer B: 10 mM $MgCl_2$, 5 mM Tris-HCl pH 8, 1 mM EDTA, and 0.05% Tween-20; pH 8

- Buffer C: 1× PBS, 1 mM EDTA, and 500 mM NaCl, pH 7.4; 0.02% Tween; optionally supplemented with 1× Trolox, 1× PCA and 1× PCD
- Blocking buffer: 1× PBS, 1 mM EDTA, 0.02% Tween-20, 0.05% NaN$_3$, 2% BSA, 0.05 mg/ml sheared salmon sperm DNA

## PCA, PCD, and Trolox

100× Trolox was made by adding 100 mg of Trolox to 430 µl of 100% methanol and 345 µl of 1 M NaOH in 3.2 ml water. 40× PCA was made by mixing 154 mg PCA in 10 ml water and NaOH and adjusting the pH to 9.0. 100× PCD was made by adding 9.3 mg PCD to 13.3 ml of buffer (100 mM Tris-HCl pH 8, 50 mM KCl, 1 mM EDTA, 50% glycerol).

## DNA origami self-assembly

All DNA origami structures were designed in Picasso Design[10]. Self-assembly of DNA origami was accomplished in a one-pot reaction mix with a total volume of 40 µl, consisting of 10 nM scaffold strands (for sequence, see Supplementary Data 1), 100 nM folding staples (Oligos named BLK in Supplementary Data 2), 250 nM biotinylated staple strands (Oligos named BIOTIN in Supplementary Data 2) and 1 µM staple strands with docking site extensions (Oligos named 5xR1 and 7xR3 in Supplementary Data 2) in 1x DNA origami folding buffer. The reaction mix was then subjected to a thermal annealing ramp using a thermocycler. First, it was incubated at 80 °C for 5 min prior to being gradually cooled from 80 to 65 °C in steps of 1 °C per 30 s and from 65 to 20 °C in steps of 1 °C per 1 min. Finally, the mix was incubated at 20 °C for 5 min and then held at 4 °C. DNA origami featuring twelve 7xR3-docking sites arranged in a 3 × 4 grid with 10 nm spacing alongside twelve 5xR1-barcode strands marking the corners of the origami (three sites per corner) and DNA origami featuring twelve 7xR3-docking sites arranged in a 3 × 4 grid with 20 nm spacing were folded.

## DNA origami purification

DNA origami structures were purified via ultrafiltration using Amicon Ultra Centrifugal Filters with a 100-kDa molecular weight cutoff (MWCO; Merck Millipore, UFC510096). The filter units were equilibrated with 500 µL of FoB5 buffer and centrifuged at 10,000 × $g$ for 5 min. Folded origamis were brought to 500 µl with FoB5 buffer and spun for 3.5 min at 10,000 $g$. This process was repeated twice. Purified DNA origami structures were recovered into a new tube by centrifugation for 5 min at 5000 × $g$ and stored at −20 °C in DNA LoBind tubes (Eppendorf, 0030108035).

## DNA-PAINT docking and imager strand sequences

The docking strands were 5xR1 (TCCTCCTCCTCCTCCTCCT), 5xR2 (ACCACCACCACCACCACCA), 7xR3 (CTCTCTCTCTCTCTCTCTC) and 7xR4 (ACACACACACACACACACA). The respective imagers were R1 (AGGAGGA-Cy3B), R2 (TGGTGGT-Cy3B), R3 (GAGAGAG-Cy3B) and R4 (7nt: TGTGTGT-Cy3B or 6nt: GTGTGT-Cy3B).

## Microscope setup

Fluorescence imaging was carried out on an inverted microscope (Nikon Instruments, Eclipse Ti2) with the Perfect Focus System, applying an objective-type TIRF configuration equipped with an oil-immersion objective (Nikon Instruments, Apo SR TIRF × 100, NA 1.49, Oil). A 560-nm laser (MPB Communications, 1 W) was used for excitation and coupled into the microscope via a Nikon manual TIRF module. The laser beam was passed through a cleanup filter (Chroma Technology, ZET561/10) and coupled into the microscope objective using a beam splitter (Chroma Technology, ZT561rdc). Fluorescence was spectrally filtered with an emission filter (Chroma Technology, ET600/50 m and ET575lp) and imaged on an sCMOS camera (Hamamatsu Fusion BT) without further magnification, resulting in an effective pixel size of 130 nm after 2 × 2 binning. 3D imaging was performed using a cylindrical lens (Nikon Instruments, N-STORM) in the detection path. Total internal reflection (TIR) or highly inclined and laminated optical sheet (HILO) illumination were performed for 2D and 3D measurements. The central 1152 × 1152 pixels (576 × 576 after binning) of the camera were used as the region of interest. The scan mode of the camera was set to "ultra quiet scan" (readout noise = 0.7 e- r.m.s., 80 µs readout time per line). Raw microscopy data was acquired using µManager (Version 2.0.1)[53].

## Single-molecule localization analysis

Raw fluorescence data were reconstructed using the Picasso software package[10] (the latest version is available at https://github.com/jungmannlab/picasso). Drift correction was performed with AIM[54]. For DNA origami an additional round of undrifting was applied by using individual 20 nm grid sites as fiducials. Crosstalk and out-of-focus localizations were filtered by the size of the images of single emitters and ellipticity. Next, two-channel datasets (DNA origami) were aligned with each other using redundant cross-correlation[55] and RESI rounds were aligned based on gold fiducials.

## DNA origami sample preparation and imaging

For sample preparation, a six-channel slide with glass bottom (Ibidi, no. 80607) was used. First, 80 µl of biotin-labeled BSA (1 mg/ml, dissolved in buffer A) was flushed into the chamber and incubated for 3 min. The chamber was then washed with 360 µl of buffer A. Then, a volume of 100 µl of neutravidin (0.1 mg/ml, dissolved in buffer A) was flushed into the chamber and incubated for 3 min. After washing with 180 µl of buffer A and subsequently with 360 µl of buffer B, 80 µl of biotin-labeled DNA structures (10 nm grid at approximately 200 pM, 20 nm grid at approximately 30 pM) in buffer B was flushed into the chamber and incubated for 5 min. After DNA origami incubation the chamber was washed 3× with 180 µl of buffer B. Finally, 2-plex Exchange-PAINT was performed by sequentially imaging the R3-grid structures followed by R1 imaging of the barcode on the 10 nm grid structure. First, 180 µl of R3 imager solution (buffer B supplemented with 1× Trolox and 600 pM R3 imager) was flushed into the chamber and imaging was performed (40,000 frames). The sample was washed three times with 1 ml of buffer B until no residual signal from the previous imager solution was detected. Then, 180 µl of R1 imager solution (buffer B supplemented with 1× Trolox and 1 nM R1 imager) was flushed into the chamber and barcode imaging was performed (10,000 frames). Imaging was performed in TIRF using 560 nm laser excitation with an exposure time of 100 ms per frame. Seven fields of view were imaged, with the R3 round acquired at laser powers ranging from 3 mW to 18 mW at the objective, corresponding to power densities between 15 and 90 W/cm$^2$ and yielding NeNA localization precisions from 2.10 nm to 5.84 nm. Barcode imaging was performed at 35 mW (175 W/cm$^2$) for all fields of views. Fresh imager solution was added for each field of view.

## Nanobody-DNA conjugation

The anti-GFP (clone 1H1, Cat No: N0305) and anti-ALFA (clone 1G5, Cat No: N1505) nanobodies were purchased from NanoTag Biotechnologies with a single ectopic cysteine at the C-terminus for site specific and quantitative conjugation. Free cysteines were reacted with 20-fold molar excess of bifunctional Sulfo DBCO-PEG4-Maleimide linker (BroadPharm, cat: BP-23318) for 2–3 h on ice. Unreacted linker was removed by buffer exchange to PBS using Amicon centrifugal filters (10,000 MWCO). 5' azide-functionalized DNA docking strand was added at 5–10 molar excess to the DBCO-nanobody and reacted overnight at 4 °C via DBCO-azide click chemistry. Unconjugated nanobody and free azide-DNA was removed by anion exchange using an ÄKTA Pure liquid chromatography system equipped with a Resource Q 1-ml column. A detailed description of the DNA conjugation to the nanobody can be found in Strauss et al. [56].

## Realistic DNA-PAINT simulations

To simulate the localizations corresponding to a labeled molecule, binding kinetics were first simulated using exponential distribution with mean defined by mean bright time (333 ms) and mean dark time (290 s). The acquisition time was capped after 1,500 s (average 20 localizations) or was proportionally increased to obtain more localizations. Next, the binding events were distributed across the frames such that they could start at any time within the exposure time, proportionally reducing the amount of collected photons within one frame. Signal that spanned less than 30% of a single frame was rejected. Molecules with <10 localizations were discarded. To simulate blinking, the amplitude of the signal was drawn from a normal distribution with mean 1000 photons and st. dev. 200 photons in a $7 \times 7$ ROI. The amplitude was then proportionally reduced to reflect the duration of the binding event within a single frame. The amplitude was used to sample a 2D Gaussian distribution simulating a single-molecule image, with sigma 0.9 pixels and background photons 100 per pixel (1 pixel = 130 nm). Then, a Poisson distribution was used to simulate camera noise with mean defined by the pixels values as per the previous step. Such blinking was then processed with Picasso: Localize using LQ 2D Gaussian fitting, obtaining the molecule's spatial coordinates and localization precision. The median localization precision obtained was 2.73 nm. Next, 2 or 12 molecules were generated with separation between them ranging from 2.73 nm to 16.38 nm in 0.273 nm steps. For each distance, 10,000 dimer simulations were conducted and 1000 grid simulations. For 3D simulations, dimers were centered at either $z = 0$ nm or $z = 260$ nm. Since axial LP varies for each z position, dimer distances from 7.8 nm to 50.0 nm and from 4.7 nm to 30.0 nm were simulated for $z = 0$ nm and $z = 260$ nm, respectively. These range in terms of median axial LP translated to the range of 1.0 LP to 6.0 LP. The calibration curve from Fig. 3 was used to extract the single-molecule image sizes at each z position. Otherwise, the same parameters as in 2D simulations above were used. After 2D localization of each simulated spot, z coordinates were calculated using the approach outlined in Huang et al. [34].

## DNA origami simulation

DNA origamis were simulated similarly to the 12 $(3 \times 4$ grid) molecule simulations described above. Since not all sites are successfully incorporated, each simulated site had a 6% probability of being rejected, as reported in the recent study[14]. Based on the same publication, it is known that the sites do not form a perfect square grid, rather, their position can be mimicked by applying Gaussian noise with $\sigma = 1.1$ nm. Since the number of localizations per origami varied for each dataset, the experimentally derived values were used - the mean number of localizations per picked origami was divided by 11.28 ( = 12 * 0.94).

## DBSCAN, ToMATo, GA and G5M parameter selection

The user parameters for DBSCAN and ToMATo were scanned to achieve the best performance and are shown in Supplementary Fig. 8. For DBSCAN, to reduce FP rate due to low localization count, at least 10 localizations were required to count a valid cluster. For GA, clustering radius of 5.46 nm was used (2 LP) and min. localizations of 10. For G5M, default sigma bounds were used (0.8 LP, 1.5 LP) and min. localizations of 10.

## DNA origami analysis

Candidate origamis were semi-manually picked with Picasso: Render's function "Pick similar". To ensure that only correctly folded origamis, the barcodes of the picked DNA origamis were analyzed: the barcode sites' centers were pinpointed with GA, then the number of barcode sites was checked to be 4, then the pairwise distances between the sites were probed—by design, the barcodes form a rectangle with sides of length 60 nm and 70 nm. Up to 5 nm deviation from ground-truth was

allowed. The sites were then visually verified for potential misfolding and rejected. DBSCAN rejected 0% to 2.4% of localizations. After the selection, G5M was used with the parameters as in the simulated case with min. localizations of 10.

## Nup96 imaging and analysis

U2OS-CRISPR-Nup96-mEGFP cells (a gift from the Ries and Ellenberg laboratories, described before[29] were cultured in McCoy's 5 A medium (Thermo Fisher Scientific, no. 16600082) supplemented with 10% FBS. Cells were passaged every 2–3 days using trypsin–EDTA. U2OS-CRISPR-Nup96-mEGFP cells were seeded on ibidi eight-well high glass-bottom chambers (no. 80807) at a density of 10,000 cm$^{-2}$. After overnight incubation, cells were fixed with 2.4% paraformaldehyde in PBS for 30 min at room temperature and washed three times with PBS. Gold nanoparticles (200 µl) were incubated for 5 min and washed three times with PBS. Blocking and permeabilization were performed with 0.25% Triton X-100 in blocking buffer for 90 min. After washing with PBS, cells were incubated with 50 nM anti-GFP nanobody (carrying the 5xR1 docking strand) in blocking buffer overnight at 4 °C. Unbound nanobodies were removed by washing three times with PBS, followed by washing once with buffer C for 10 min. Postfixation was performed with 2.4% paraformaldehyde in PBS for 7 min. After washing $3\times$ with PBS, the imager solution (buffer C supplemented with $1\times$ Trolox, $1\times$ PCA and $1\times$ PCD and 80 pM R1 imager) was flushed into the chamber. 130,000 frames were acquired at 100 ms exposure time in HILO mode using 20 mW of 560 laser excitation (100 W/cm$^2$). Image was analyzed with Picasso as described above. DBSCAN removed 14% of localizations. Upon applying G5M (min. localizations = 15, default σ bounds) to localizations, the alignment with LocMoFit was conducted as described earlier[13].

## CD20 imaging and analysis

A subset of CHO-K1 mEGFP-CD20 cells from Pachmayr et al. [30] was reanalyzed for this publication. CHO-K1 (CCL-61, ATCC) cells were cultured in Gibco Ham's F-12K (Kaighn's) medium (Thermo Fisher Scientific, cat: 21127030), supplemented with 10% FBS (cat. A5669701, Gibco) and seeded in ibidi eight-well high glass-bottom chambers (10,000 cells per well) the day prior to transfection. Cells were transfected with a mEGFP-CD20[30] construct using Lipofectamine LTX and allowed to express overnight. Cells were treated for 30 min at 37 °C with 66.7 nM of Obinutuzumab or CD20-CD3 T cell engagers (classical or inverted) diluted in medium. Therapeutic antibodies were provided by Roche Glycart. After washing twice with F-12K medium, cells were fixed with prewarmed 4% PFA for 15 min and subsequently washed with PBS. Cells were permeabilized with 0.1% TritonX-100 for 5 min and blocked for 1 h at RT with blocking buffer. The RESI staining mix was prepared by pre-mixing anti-GFP nanobodies conjugated with DNA-docking strands 5xR1, 5xR2, 7xR3, and 7xR4 in equimolar amounts at a final concentration of 50 nM in blocking buffer and incubated overnight at 4 °C. Prior to imaging, unbound nanobodies were washed away, the sample was postfixed with 4% PFA + 0.1% glutaraldehyde in PBS for 10 min and quenched with 0.2 M NH$_4$Cl in PBS. Gold nanoparticles were diluted 1:3 in PBS and incubated for 7 min. After washing, four rounds of RESI imaging were acquired sequentially with 500 pM to 1 nM of R1, R2, R3 and R4 imagers using a laser power of 30 mW at the objective (150 W/cm$^2$). Raw data was analyzed with the standard Picasso workflow described above. DBSCAN rejected 4% to 15% of localizations. Protein positions were identified by GA and G5M with min. localizations set to 15 for both algorithms. DNA-PAINT datasets were obtained by merging DNA-PAINT localizations from the four RESI rounds prior to GA and G5M analysis. CD20 oligomer stoichiometries were obtained using SPINNA[14] with label uncertainty of 5 nm, labeling efficiency of 47%, granularity of 51 and number of simulations of 50. The model oligomers were as follows: monomer, dimer of intermolecular distance of 12 nm, trimer— equilateral triangle

of side length 12 nm and tetramer−square of the same side length. To ensure accurate fitting around the 2nd and 3rd NND peaks, SPINNA was modified to assign higher weights to lower distances (Gaussian distribution with mean at 12, 18, and 20 nm for 1st, 2nd and 3rd NNDs, respectively and σ of 16 nm) and Wasserstein distance was used rather than Kolmogorov-Smirnov for scoring model fitness.

## PD-L1 imaging

For 2D imaging, 3000 CHO-K1 (CCL-61, ATCC) were seeded in a well on a coverslip (Ibidi, Cat.No: 80827) and cultured for 24 h at 37 °C in Gibco Ham's F-12K (Kaighn's) medium (Thermo Fisher Scientific, cat: 21127030). The cells were transfected with an overexpression construct (PD-L1-ALFA-mEGFP[47] using Lipofectamine™ 3000 as specified by the manufacturer (Invitrogen™, Cat.No: L3000008) for 2 days. For 3D PD-L1 imaging, the PD-L1-mEGFP CHO cell line was a gift from the Ian Parish Lab. 30,000 CHO cells were seeded in a well on a coverslip (Ibidi, Cat.No: 80827) and cultured for 24 h at 37 °C. Subsequent procedures were all carried out at room temperature and pressure. The cells were fixed for 15 min with 4% PFA prewarmed to 37 °C, permeabilized with 0.1% Triton X-100 for 5 min and blocked for 60 min with a blocking buffer. For 2D PD-L1 imaging, the cells were stained with anti-ALFA at 25 nM in a blocking buffer for 60 min. For 3D PD-L1 imaging, the cells were stained with anti-GFP nanobodies at 50 nM in a blocking buffer for 60 min. Both nanobodies were conjugated with the 7xR4 sequence. Three washing steps with PBS were carried out between each step. The sample was incubated in Buffer C for 5 min and then postfixed with 2% PFA + 0.2% glutaraldehyde for 30 min and quenched with 100 mM $NH_4Cl$ for 20 min. Before imaging, gold nanoparticles were added to the sample for 5 min and the imaging buffer was added (6 nt R4 imager for 2D, R4 for 3D). TIRF imaging with a 561 nm excitation laser was conducted with a power of 25 mW at the objective (125 W/cm$^2$). 2D imaging was acquired across 30,000 frames with 75 ms integration time and 3D across 50,000 frames with 100 ms integration time.

## Normality tests

Individual molecules were manually picked in the regions without neighboring molecules. Circular picks with a diameter of 26 nm were chosen to envelop all localizations likely originating from the picked molecules. To ensure that no molecules were overlapping in the axial dimension in the 3D dataset, each pick was visually checked for such an overlap.

## Reporting summary

Further information on research design is available in the Nature Portfolio Reporting Summary linked to this article.

## Data availability

The single-molecule localization data generated in this study have been deposited in the Zenodo database at https://doi.org/10.5281/zenodo.18429674. Source data are provided with this paper.

## Code availability

Raw image processing and G5M can be performed using Picasso v0.9.5 available via GitHub at https://github.com/jungmannlab/picasso and https://doi.org/10.5281/zenodo.18416173. Custom analysis scripts are available via Zenodo at https://doi.org/10.5281/zenodo.18429674.

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

## Acknowledgements

We are thankful to Monique Honsa, Philipp R. Steen, Ondřej Skořepa and Eva-Maria Schentarra for testing the software and providing feedback. R.K., S.C.M.R., I.P., and S.X. acknowledge the support by the IMPRS-ML graduate school. L.A.M. acknowledges the postdoctoral fellowship from the European Union's Horizon 20212022 research and innovation program under Marie Skłodowska-Curie grant agreement no. 101065980. This research was funded in part by the European Research Council through an ERC Consolidator Grant (ReceptorPAINT, grant agreement number 101003275), the BMBF (Project IMAGINE, FKZ: 13N15990), the Volkswagen Foundation through the initiative 'Life?—A Fresh Scientific Approach to the Basic Principles of Life' (grant no. 98198), the Danish National Research Foundation (Centre for Cellular Signal Patterns, DNRF135), the Max Planck Foundation and the Max Planck Society.

## Author contributions

R.K. conceived and developed the algorithms, implemented the software, and analyzed the data, S.C.M.R. contributed to developing the algorithms, performed experiments, and analyzed data, I.P. and S.X. performed experiments, R.K., S.C.M.R., L.A.M., and R.J. interpreted data. R.K., S.C.M.R., L.A.M., and R.J. wrote the manuscript with input from all authors, L.A.M. and R.J. supervised the project. R.K. and S.C.M.R. contributed equally.

## Funding

## Competing interests

The authors declare no competing interests.

## Additional information

**Peerreview information** *Nature Communications* thanks Jesse Goyette, who co-reviewed with Shirin Ansari; Xianan Qin and the other, anonymous, reviewer(s) for their contribution to the peer review of this work. A peer review file is available.

