## [Transparent Peer Review file · Nature Communications]

Molecular mapping in DNA-PAINT via modified Gaussian Mixture Modeling

Corresponding Author: Professor Ralf Jungmann

Version 0:

Reviewer comments:

Reviewer #1

(Remarks to the Author)

In this manuscript, Kowalewski et al report an expectation-maximization (EM) algorithm based method to resolve localization information from fluorescently labeled molecular clusters (G5M) and demonstrate superiority of G5M over existing methods. The manuscript is well written: the demonstrated performance of G5M is for sure a great improvement over the existing methods (28-fold higher recovery rate than current methods and <0.1% false positives, on simulation data). This work is methodologically sound. In principle, I think that this work is worth publishing in Nature communications.

Below are my comments. Hopefully, the authors may consider about them and include some of them in the future versions of their manuscript.

(1) The authors demonstrate that the existing definition for Rayleigh limit is not suitable to SMLM. Their writing "By translating the Rayleigh criterion to the signal produced by two gaussian distributions (Fig. S2)" is somewhat confusing. The new definition is actually "produced" by "fitting Gaussians to Bessels" but not from "two gaussians". Also, I personally think that the new definition is also "general" to general optics and microscopy, as long as the apertures of the optical system are round shape -> which produces "Bessel" Airy disks. The authors might revise their description for the definitions of diffraction limit in a more accurate way -> the above are based on my understanding and just goes to a coefficient to the localization precision (LP) and does not affect to much to the core of their conclusions. Anyway, I suggest to make it clear in the manuscript.

(2) The math description for G5M is somewhat concise. Anyhow, I think it is better to make it detailed, because it readily increase the readability and reproducibility of your work. In general EM algorithm for GMM, the algorithm is actually dealing with "probability". What is the "probability" p defined in this work? (the probability of finding a location belongs to a given cluster? or the probability of other definitions?) The definitions here are important for readers to follow. Also, what is the "GMM" and "X" in the brackets of your "Mathematical foundation and Expectation-Maximization's (EM) overview" in the supporting information? My suggestion is to rewrite this part in the supporting information and make it clear.

(3) EM algorithm requires the initiative parameters of "number of components". For simulation data and artificial data such as fluorescently labeled grids, this information is clear. But for many practical cases, this is hard to determine. Is a single metric BIC enough to determine this important information? Some discussion might be needed. (The current version of discussion is more like a conclusion section and lacks discussion on these essential aspects). Also, how does G5M assigns the "labels" in the first step of the EM algorithm? (by conventional K-Means as provided by the GMM in the Sklearn package?).

(4) As the authors demonstrated, G5M can be applied to the 1D, 2D and 3D cases. But somehow the application on the simulation data and the artificially labeled grids are only on the 2D case. Is it possible to include the 3D case (at least for the simulation data)? I find Panel f of Figure 1 is related to this context, but somehow it lacks in the Figure 2.

(5) EM algorithm can be applied to mixtures of other probability functions, but not just Gaussian mixtures. So the current work is based on a understanding that the localized points are mixed Gaussians. Is it fundamentally correct? Is it possible if Gaussian is changed into other proper functions, leading to improvement of performance? (of course the identifiability of the new mixtures should be considered). The authors might discuss on this.

(Remarks on code availability)

Reviewer #2

(Remarks to the Author)

Kowalewski and colleagues propose another method to analyze SMLM data, adapted to DNA-PAINT labelling technique and

based on Gaussian Mixture Modeling. G5M method considers as priors the localization precision, spatial constraints, and DNA hybridization kinetics.

SMLM are powerful super-resolution imaging methods but they suffer from limited tools to exploit the meaning of their data. In this sense, a new method is always welcomed. And in case of DNA-PAINT, this is even more helpful giving the fact that in the literature most developments target PALM and STORM methods.

Nevertheless, I have some reserves in cautioning G5M. Essentially, I do not see how useful could be to use it.

My main points:

1) I had a hard time to understand the logic of G5M to find the “clouds of points” that are expected to belong to one molecule, neglecting time completely.

At least some of the algorithms currently used in labs to detect SMLM provide a tool to correct multiple observations of the same fluorophore. This is similar to what G5M does... but in a much simpler way, I think, applicable to DNA-PAINT (in which a fluorophore binds and unbinds rapidly to the target molecule). Briefly, the idea is that all the detections of the same molecule will be grouped in time. So it is just about collecting the detections that are consecutive in time, in the area expected given the localization precision (and the molecule size). Could the authors explain why they do not consider time? is it a problem of computational burden?

2) Indeed, using time as a clustering method could replace DBSCAN in conditions of sparse labeling. DBSCAN has serious problems with background noise and spurious clustering and authors mention the problem of false negatives as well. Could the authors at least provide some cues about how to set the minimum number of detections for DBSCAN, knowing that this may change from one experiment to the other?

3) The ideal number of detections per molecule for G5M to work properly seems to be in a very narrow range (around 20) and changing the parameters does not improve the yield. If this is the case, it should be more clearly stated and explained.

4) The method looks nice with the examples provided (with the good number of detections and distances between molecules), but I'm convinced that G5M will not necessarily work so nicely on all kinds of datasets. This is very easy to test using synthetic data. The authors explored this somehow (Fig 2) but they show only two situations thus it is not clear for me, as a potential user, whether this method would adapt to my data.

5) In the abstract, the claim that “G5M resolves molecules at the Rayleigh limit with a 28-fold higher recovery rate” is true only for some conditions as the number of detections is determinant for the performance of G5M in Fig. 2D. This assertion should be tempered by telling in which conditions this result was obtained.

6) Also in the abstract, the result “<0.1% false positives” could be determined on chosen synthetic data. I guess that in different conditions this value is not true. It would be very helpful to discuss and clarify the limits of applicability of the method showing in which conditions it will not work so perfectly. There are a couple of supplemental figures that show limitations, but they are not discussed at all.

7) Again in the abstract, I have an epistemological issue with the sentence “Applied to experimental datasets, G5M recovers full nuclear pore complex structures and detects higher-order CD20 oligomers induced by antibody treatment, outperforming conventional DNA-PAINT analysis”. In this case, you do not know the ground truth, so how can you be sure that your observation is 100% real and it is not an artifact of your method? G5M could correctly group more detections, but how can you be sure that these detections correctly reflect the number of molecules?

Other points:

- It was not clear for me how to navigate through supplemental material. There is one file called “Supplementary Information”, two files “Supplementary Data” (with raw DNA data, no legend, no title), four “Supplementary Tables” without legends. Could you create a unique file with all this, with all items well described in legends?

- Data in supplemental tables are impossible to understand. I could not find legends, and tables do not show the borders to separate columns. For example, in supplemental table 1 it reads “n localizations GA (s)”. Clearly, there is a formatting problem...

- The interest of the information provided by supplemental fig. S1 is not clear. For me, by definition, detections distribute following a normal distribution around the position of the molecule. I do not understand why you need to prove this.

- Fig. S3 is barely explained with respect to parameters values.

- The reference to Fig. 4b in page 4 is not correct (should be 4c-d).

- Which is the interest of Fig. S5? Figs S11-14 are also useless as they are not discussed.

As a conclusion I see a new method that could be a little faster (depending on the number of detections) and more efficient than GA in a narrow range of situations or experiments. Sadly, its efficiency cannot be improved, as changing parameters do not change the result significantly. If I am wrong, please show it to me.

(Remarks on code availability)

Reviewer #3

(Remarks to the Author)

The manuscript introduces G5M, a probabilistic algorithm designed to extract molecular positions from DNA-PAINT single-molecule localisation microscopy data. The authors benchmark G5M against simulations of different molecular organisations, showing significant improvement over their previous gradient ascent (GA) methodology. Validation on simulated datasets, DNA origami structures, nuclear pore complexes, and CD20 receptor oligomers demonstrates that G5M reliably recovers known molecular architectures and reveals higher-order assemblies that are missed by the current GA approach.

By embedding G5M within the open-source Picasso platform, the authors make a sophisticated modelling framework broadly accessible to the super-resolution community. The anticipated impact of this work for the PAINT imaging community is significant: it provides a generalizable, statistically principled method for molecular mapping that enhances both the accuracy and interpretability of PAINT data. How widely this may be applied to SMLM data from methods other than DNA-PAINT remains to be seen, but the work is a strong addition to the DNA-PAINT analysis framework. I have only a few very technical reservations, which I have outlined below.

Points of clarification:

The assumption that localisations are normally distributed may be reasonable for ideal datasets (e.g., minimal drift, uniform TIRF illumination). However, if linked localisations are used, the precision of the linked molecules may not follow a Gaussian distribution, since position averaging during linking will typically increase precision. How would this impact on the performance of the G5M algorithm? This should be briefly discussed to inform users of appropriate input data format.

It was not clear from the manuscript how localisation precision is defined. The provided code also does not prompt the user for this value, and it is unclear where it is extracted from. Is the localisation precision defined as the average fitting precision of individual localisations, or as the NeNA localisation precision? If the latter, the following reference should be cited: Endesfelder et al. *Histochem. Cell Biol.* 141, 629–638 (2014). This citation should also be added in the “DNA origami sample preparation and imaging” section of the Methods (page 15).

Molecular position precision in different conditions:

Please include a metric for the nearest-neighbour distance between the identified molecular positions and the ground truth positions in the simulated data (Figure 2). This would complement the reported relative detection numbers (Figures 2d and 2h) and provide a clearer measure of the G5M algorithm's performance. It would also be valuable to examine how the precision of molecular positioning depends on the average number of localisations collected per molecule.

Dependence on pre-clustering with DBSCAN:

As noted in the Supplementary Material (page 5), the performance of the G5M algorithm is highly dependent on the quality of the initial clustering. This represents a critical point where variability in data will necessitate user expertise in parameter selection. Page 8 encourages users to experiment with the DBSCAN parameters, but there is a lack of objective measurements to provide guidance on what constitutes a successful or unsuccessful clustering outcome.

The manuscript suggests that DBSCAN should be tuned to (i) remove background and retain “true” localisations, and (ii) divide the data into subsets containing fewer than 10 molecules. This advice assumes that users can reliably identify “true” localisations and know the approximate number of molecules contributing to each cluster; but this is precisely the quantities that the G5M algorithm is intended to infer. Consequently, this step introduces a high degree of subjectivity into the analysis. Perhaps the authors could provide a simple tool to help users assess the DBSCAN clustering output. For example, they might include a function that quantifies the area of clusters produced by a user's defined parameter settings and reports the proportion of clusters that exceed a benchmark size. This could be the area corresponding to the tightest distribution of localisations that the G5M algorithm could fit ten molecules to, such as the combined area of ten circles with radii equal to half the Rayleigh limit (based on the given localisation precision), or another similarly justified threshold.

A complementary diagnostic to evaluate background removal would also be valuable. For example, the authors could report the proportion of localisations classified as “noise” by DBSCAN for given parameters, or visualise how this fraction varies as parameters are adjusted. Additionally, an objective performance measure could be obtained by reconstructing images from (i) all localisations and (ii) only those retained as clustered, using pixel bins approximately half the localisation precision, and then comparing their Fourier Ring Correlation (FRC) resolutions. An improvement in FRC resolution for the clustered subset would indicate that background removal has produced a sharper and more spatially coherent signal. While FRC is not a direct measure of noise suppression, it would provide users with a reproducible, quantitative means of assessing whether their parameter choices have removed background and enhanced data quality.

Additional validation metric:

It would also be useful to measure the influx rate within each DBSCAN cluster. Given known on- and off-rates for binding, this could provide an independent estimate of the number of molecules per cluster. Comparing this to the number of sites fitted by G5M would offer an indicator of underfitting (e.g., missing closely spaced sites), particularly when applied to experimental data.

(Remarks on code availability)

The code was relatively easy to install and implement in the Picasso environment. I particularly found parameterising the DBSCAN a little challenging and difficult to assess when I tried it on some data from my lab. The lack of GUI for DBSCAN results and lack of definition of localisation precision made it feel like a substantial proportion of the process was buried "under the hood".

Reviewer #4

(Remarks to the Author)

(Remarks on code availability)

Version 1:

Reviewer comments:

Reviewer #1

(Remarks to the Author)

I think that the authors have well addressed my concerns. The manuscript has been considerably improved. I recommend publication of this manuscript in its current form.

(Remarks on code availability)

Reviewer #2

(Remarks to the Author)

The authors addressed all my concerns, answered all my questions and modified the manuscript when needed. I thus recommend the article for publication.

(Remarks on code availability)

Reviewer #3

(Remarks to the Author)

I would like to thank the authors for engaging sincerely with my feedback. All of my comments have been addressed.

(Remarks on code availability)

Revisions have helped clarify use, particularly for experienced Picasso users.

Reviewer #4

(Remarks to the Author)

(Remarks on code availability)

The read me file was detailed enough and easy to follow to allow integration of the G5M code into the existing picasso environment. The example localised hdf5 files provided in the dropbox link above were well labelled so that it was easy to understand how the data was processed through the G5M pipeline.

REVIEWER COMMENTS

We thank all reviewers for their suggestions and comments. As a result, we adjusted the DBSCAN parameters and modified 3D G5M. This improved the results in the figures which now have been updated.

Reviewer #1 (Remarks to the Author):

In this manuscript, Kowalewski et al report an expectation-maximization (EM) algorithm based method to resolve localization information from fluorescently labeled molecular clusters (G5M) and demonstrate superiority of G5M over existing methods. The manuscript is well written: the demonstrated performance of G5M is for sure a great improvement over the existing methods (28-fold higher recovery rate than current methods and <0.1% false positives, on simulation data). This work is methodologically sound. In principle, I think that this work is worth publishing in Nature communications.

Below are my comments. Hopefully, the authors may consider about them and include some of them in the future versions of their manuscript.

We thank the reviewer for their appreciation of our work.

(1) The authors demonstrate that the existing definition for Rayleigh limit is not suitable to SMLM. Their writing "By translating the Rayleigh criterion to the signal produced by two gaussian distributions (Fig. S2)" is somewhat confusing. The new definition is actually "produced" by "fitting Gaussians to Bessels" but not from "two gaussians". Also, I personally think that the new definition is also "general" to general optics and microscopy, as long as the apertures of the optical system are round shape-> which produces "Bessel" Airy disks. The authors might revise their description for the definitions of diffraction limit in a more accurate way.-> the above are based on my understanding and just goes to a coefficient to the localization precision (LP) and does not affect to much to the core of their conclusions. Anyway, I suggest to make it clear in the manuscript.

We have now clarified this sentence in the main text and added a section "**2 Rayleigh Criterion for Localizations**" in the SI explaining the details of our definition.

(2) The math description for G5M is somewhat concise. Anyhow, I think it is better to make it detailed, because it readily increase the readability and reproducibility of your work. In general EM algorithm for GMM, the algorithm is actually dealing with "probability". What is the "probability" p defined in this work? (the probability of finding a location belongs to a given cluster? or the probability of other definitions?) The definitions here are important for readers to follow. Also, what is the "GMM" and "X" in the brackets of your "Mathematical foundation and Expectation-Maximization's (EM) overview" in the supporting information? My suggestion is to rewrite this part in the supporting information and make it clear.

We have significantly extended the sections "**3.1 Mathematical foundation and Expectation-Maximization's (EM) overview**" and **3.3.2 Log-likelihood filtering** in the Supplementary Information. It now provides a detailed, step-by-step explanation of EM with Gaussian Mixture Models.

Specifically:

X describes the coordinates of N localizations ($X=(x_1, \dots, x_n, \dots, x_N)$ with x_n being a two or three dimensional vector).

GMM stands for a given Gaussian Mixture Model with K components and specific mean values, covariance matrices and weights $\{(\mu_1, \Sigma_1, \pi_1), \dots, (\mu_K, \Sigma_K, \pi_K)\}$. In agreement with standard nomenclature, we now changed the name of the parameters of the GMM to θ to improve clarity ($\theta=\{(\mu_1, \Sigma_1, \pi_1), \dots, (\mu_K, \Sigma_K, \pi_K)\}$).

The probability $p(X|\theta)$ (formerly $p(X|GMM)$) describes the probability that a dataset X is generated by the GMM with parameters $\theta = \{(\mu_1, \Sigma_1, \pi_1), \dots, (\mu_K, \Sigma_K, \pi_K)\}$. The same expression can be interpreted as the likelihood $L(\theta|X)$ (formerly $L(GMM|X)$) of the parameters θ of the GMM given that the dataset X was observed.

(3) EM algorithm requires the initiative parameters of “number of components”. For simulation data and artificial data such as fluorescently labeled grids, this information is clear. But for many practical cases, this is hard to determine. Is a single metric BIC enough to determine this important information? Some discussion might be needed. (The current version of discussion is more like a conclusion section and lacks discussion on these essential aspects).

We thank the reviewer for this remark. BIC is known to be asymptotically optimal, thus it will identify the correct model (if it is one of the candidate models!) in the extreme case of $n \rightarrow \infty$, where n is the number of datapoints (localizations). We have observed that BIC alone would not be sufficient to determine the correct model (correct number of components). However, G5M includes other steps in the algorithm (σ and N_{locs} criteria) to enhance the accuracy of the model determination.

Altogether, in *in silico* data and DNA origami, G5M identifies the correct number of components reproducing the known number of sites. This demonstrates that G5M is accurate for as few as $n = 20$ localizations per molecule. In Figure S3 we show the role of excluding models with components validating the min. and max. σ or the min. number of localizations N_{locs} criteria.

Moreover, our cellular measurements of Nup96 correctly identify protein dimers in good agreement with previously reported structural biology data. Only in 1.6% of cases did we find larger structures of more than two Nup96 molecules. This confirms that the combination of BIC with the σ and N_{locs} criteria correctly identifies the number of components for the GMM. We have now discussed this in more detail in the paper and added information to Table S4.

Also, how does G5M assigns the “labels” in the first step of the EM algorithm? (by conventional K-Means as provided by the GMM in the Sklearn package?).

G5M assigns the initial labels using k-means++. We have extended the explanation of the initialization in **Section 3a** in the Supplementary Information.

(4) As the authors demonstrated, G5M can be applied to the 1D, 2D and 3D cases. But somehow the application on the simulation data and the artificially labeled grids are only on the 2D case. Is it possible to include the 3D case (at least for the simulation data)? I find Panel f of Figure 1 is related to this context, but somehow it lacks in the Figure 2.

We thank the reviewer for this valuable suggestion. We have now added **Figure S12** that compares the performance of GA and G5M on two sets of dimer simulations - one at $z = 0$ where spot width and height are roughly equal and one at $z = 260$ nm where spot width is significantly larger than its height. In both cases, G5M outperforms GA.

Moreover, we decided to reevaluate G5M’s mechanism in 3D. We have improved the choice of Gaussian component’s σ bounds by introducing a theoretical formula for axial localization precision. The derivation can be found in **Section 4 Axial localization precision** in the Supplementary Information.

(5) EM algorithm can be applied to mixtures of other probability functions, but not just Gaussian mixtures. So the current work is based on a understanding that the localized points are mixed Gaussians. Is it fundamentally correct? Is it possible if Gaussian is changed into other proper functions, leading to improvement of performance? (of course the identifiability of the new mixtures should be considered). The authors might discuss on this.

In the context of widefield imaging of fluorophores that behave as freely rotating dipoles, each localization is an estimation of the expected value of the Point Spread Function (PSF) of the microscope. Thus, localizations must follow a Gaussian distribution according to the Central Limit Theorem. Therefore, we argue that it is fundamentally correct to state that localizations follow a normal distribution.

We further confirm this theoretical prediction with our experimental measurements (Fig. S1), demonstrating the absence of artifacts, such as drift, that might otherwise alter the expected Gaussian distribution. We added a dedicated section discussing this in detail in the Supplementary Information (see **Section 1 Single-Molecule Localizations are Gaussian distributed**). Because of this reason, a mixture of Gaussians is the appropriate choice, and using a mixture of other distributions is likely incorrect.

Reviewer #2 (Remarks to the Author):

Kowalewski and colleagues propose another method to analyze SMLM data, adapted to DNA-PAINT labelling technique and based on Gaussian Mixture Modeling. G5M method considers as priors the localization precision, spatial constraints, and DNA hybridization kinetics.

SMLM are powerful super-resolution imaging methods but they suffer from limited tools to exploit the meaning of their data. In this sense, a new method is always welcomed. And in case of DNA-PAINT, this is even more helpful giving the fact that in the literature most developments target PALM and STORM methods.

Nevertheless, I have some reserves in cautioning G5M. Essentially, I do not see how useful could be to use it.

My main points:

1) I had a hard time to understand the logic of G5M to find the “clouds of points” that are expected to belong to one molecule, neglecting time completely.

At least some of the algorithms currently used in labs to detect SMLM provide a tool to correct multiple observations of the same fluorophore. This is similar to what G5M does... but in a much simpler way, I think, applicable to DNA-PAINT (in which a fluorophore binds and unbinds rapidly to the target molecule). Briefly, the idea is that all the detections of the same molecule will be grouped in time. So it is just about collecting the detections that are consecutive in time, in the area expected given the localization precision (and the molecule size). Could the authors explain why they do not consider time? is it a problem of computational burden?

The reviewer is right in mentioning that multiple algorithms have been developed to correct for multiple observations of the same fluorophore; in fact, our own software, Picasso, provides an implementation of such an algorithm. However, this is not the goal of G5M, which is fundamentally different in scope and content. Briefly, G5M aims at inferring the true position of the target biomolecule from the set of localizations stemming from many different binding events of different fluorophores throughout the measurement.

In DNA-PAINT, multiple localizations are obtained from transiently binding events of different fluorophore-conjugated single-stranded DNAs (“imagers”) to the same “docking” single-stranded DNA sequence (see **Figure R1**). A binding event can span over several camera frames; thus, the localizations from the same binding event of an imager to a docking strand are correlated in time. However, the timepoints of the next hybridization events are independent of the timepoints of the previous hybridizations, thus not providing any useful information to group localizations coming from the same target biomolecule.

The localizations obtained from all binding events (each event corresponding to a different imager) to one docking strand form a Gaussian distribution (a “cloud” of points) around the target biomolecule's position with a spread matching the localization precision (typically 2-5 nm in DNA-PAINT). G5M aims to fully harness this spatial information to infer the true position of the target biomolecule (usually a protein). This makes the use of the time information to group localizations into molecule coordinates not essential for our method.

Figure R1. Top: Localizations and time trace of a single DNA-docking strand in a DNA-PAINT experiment lasting 40000 frames. Each “ON” spike corresponds to one binding event of an imager to the docking strand. **Bottom:** Zoom in to the binding event highlighted in red in the top panel. The imager is bound for four frames, producing four localizations that are correlated in time.

2) Indeed, using time as a clustering method could replace DBSCAN in conditions of sparse labeling. DBSCAN has serious problems with background noise and spurious clustering and authors mention the problem of false negatives as well. Could the authors at least provide some cues about how to set the minimum number of detections for DBSCAN, knowing that this may change from one experiment to the other?

We politely disagree with the reviewer as using time as clustering cannot replace DBSCAN for the reasons stated in our previous answer.

Following the suggestion of the reviewer, we have now added extra information and advice on how to set the DBSCAN parameters to optimize the results of G5M (**Supplementary Information section 3.4** and **Fig. S8**).

3) The ideal number of detections per molecule for G5M to work properly seems to be in a very narrow range (around 20) and changing the parameters does not improve the yield. If this is the case, it should be more clearly stated and explained.

In Figure 2d, we show that the recovery rate for true positive detections improves when more localizations per molecule are available. The maximum recovery rate for true positives improves from 85% for an average of 20 localizations per molecule to 94% for 30 localizations and 97% for 40 localizations (see Supplementary Table 2 for fit results of data in Fig. 2d). In summary, we provide minimum numbers of localizations for given target fidelities but, in general, the larger the number of localizations the better G5M will perform.

4) The method looks nice with the examples provided (with the good number of detections and distances between molecules), but I’m convinced that G5M will not necessarily work so nicely on all kinds of datasets. This is very easy to test using synthetic data. The authors explored this somehow (Fig 2) but they show only two situations thus it is not clear for me, as a potential user, whether this method would adapt to my data.

We show that G5M successfully identifies molecule coordinates in DNA origami (Fig 2i-j) as well as in the cellular environment by imaging the Nuclear Pore Complexes and CD20 (Fig. 3 and Fig. 4). However, it is correct that G5M is not applicable to all kinds of datasets, as we specifically tailored it to typical DNA-PAINT

data. Figure 2a-h shows general and quantitative minimal requirements on the data for it to be suitable for G5M analysis, e.g. a minimum of 20 localizations per molecule on average and distances at least 3 times larger than the localization precision (LP) for simple structures like dimers. For more complex and dense data like the 3x4 grids, distances larger than 4.5 x LP can be resolved accurately if 30 or more localizations were acquired per molecule (Fig. 2e-h). Data of lower quality (see Fig S6) does not provide enough spatial information to measure molecule coordinates with the high precision needed for any downstream analysis of spatial arrangements of molecule complexes.

5) In the abstract, the claim that “G5M resolves molecules at the Rayleigh limit with a 28-fold higher recovery rate” is true only for some conditions as the number of detections is determinant for the performance of G5M in Fig. 2D. This assertion should be tempered by telling in which conditions this result was obtained.

We thank the reviewer for this remark. We now specify the conditions in the abstract and the introduction.

6) Also in the abstract, the result “<0.1% false positives” could be determined on chosen synthetic data. I guess that in different conditions this value is not true. It would be very helpful to discuss and clarify the limits of applicability of the method showing in which conditions it will not work so perfectly. There are a couple of supplemental figures that show limitations, but they are not discussed at all.

The conditions on which the “<0.1% false positives” claim is based are now specified in the abstract and the introduction. Indeed, the false positive rate can change with the conditions: The <0.1% false positives were obtained for simulations of dimers with 20 localizations on average per molecule (**Fig. 2a-d**), whereas the simulation of the 3x4 grid with 30 localizations on average per molecule contained 0.5% false positives (**Fig. 2e-g**). However, we note that the percentage of false positives is consistently low across distances between molecules ranging from 1xLP to 6xLP, while also being robust to changes of user parameters, thus preventing the introduction of artifacts by G5M analysis (**Fig. S9**). Moreover, Picasso routinely performs the test demonstrated in **Figure S15** when running G5M. We added a remark to the text.

7) Again in the abstract, I have an epistemological issue with the sentence “Applied to experimental datasets, G5M recovers full nuclear pore complex structures and detects higher-order CD20 oligomers induced by antibody treatment, outperforming conventional DNA-PAINT analysis”. In this case, you do not know the ground truth, so how can you be sure that your observation is 100% real and it is not an artifact of your method? G5M could correctly group more detections, but how can you be sure that these detections correctly reflect the number of molecules?

In these experimental datasets, a perfect ground truth does not exist, indeed. We do not know how many docking strands underlie a specific group of localizations. However, cryo-EM and RESI data provide a higher-resolution reference, which is a very good approximation to a ground truth. For example, in the case of NPC, we do know that two Nup96 molecules are expected per symmetry site on each ring. With our G5M analysis, we found that 98.4% of detected, filled NPC segments contain one or two, but not more, NPC molecules (Table 4) in agreement with cryo-EM. This strengthens the conclusion from *in silico* and DNA origami data (where ground truth exists) that G5M accurately assigns the number of Gaussians to the localizations. We added this information now explicitly to the Results and Table 4 and discussed it in the paper.

Other points:

- It was not clear for me how to navigate through supplemental material. There is one file called “Supplementary Information”, two files “Supplementary Data” (with raw DNA data, no legend, no title), four “Supplementary Tables” without legends. Could you create a unique file with all this, with all items well described in legends?

Our supplementary data and tables are arranged to follow journal guidelines, allowing for the Submission of these as individual files with titles. Their content is described when the respective supplementary material is referenced in the text. We agree with the Reviewer that the clarity can be improved. To do so, we have now

extended the titles in the filenames. Moreover, in the methods section, when describing the different types of DNA strands used during DNA origami self-assembly, we now mention the exact names under which they are listed in Supplementary Data 2. We also added information about the data and the computer used for the runtime tests within Table 1.

- Data in supplemental tables are impossible to understand. I could not find legends, and tables do not show the borders to separate columns. For example, in supplemental table 1 it reads “n localizations GA (s)”. Clearly, there is a formatting problem...

Most likely the pdf conversion within the manuscript tracking system formatted the tables. When reading in Microsoft Excel, the tables should be easy to read, see the screenshot below. For clarity, we have now added explicit borders separating columns and rows to improve readability.

Moreover, we renamed the column headers in Table S1:

- “n localizations” → “Dataset size (n localizations)”
- “GA (s)” → “Runtime GA (s)”
- “G5M (s)” → “Runtime G5M (s)”

5	Dataset size (n localizations)	Runtime GA (s)	Runtime G5M (s)
6	286436	35	89
7	536371	65	104
8	1209359	153	161
9	2276081	263	248
10	4381904	539	306
11	7095431	928	506

- The interest of the information provided by supplemental fig. S1 is not clear. For me, by definition, detections distribute following a normal distribution around the position of the molecule. I do not understand why you need to prove this.

We agree that, theoretically, the single-molecule detections should follow a Gaussian distribution around the true molecule position. However, given that this is a central and fundamental result for our work, i.e., the importance of choosing the right distribution for the mixture model, we confirmed this theoretically expected result with our experimental data. We note that, for example, an unsuccessful correction of the thermal/mechanical drift of the sample during the measurement could very well lead to non-Gaussian distributions of localizations. Therefore, in **Fig. S1**, we demonstrate that indeed, the localizations under our experimental conditions follow a normal distribution.

- Fig. S3 is barely explained with respect to parameters values.

We thank the reviewer for the valid remark. We have now added the used parameters explicitly in the caption of Fig. S3. They are the recommended values also specified in the main text and the Methods.

- The reference to Fig. 4b in page 4 is not correct (should be 4c-d).

We thank the reviewer for pointing this out. The references are now corrected for **Fig. S4**.

- Which is the interest of Fig. S5? Figs S11-14 are also useless as they are not discussed.

Fig. S5 measures the effect of different numbers of localizations acquired per molecule on G5M performance. It extends Fig. 2d, which shows the % of true positives for an average of 20, 30, and 40 localizations per molecule by quantifying the respective false negatives and false positives. We now refer to **Fig. S11-14** (now numbered as **Fig. S14-S17**) individually in the text and explain their purpose.

As a conclusion I see a new method that could be a little faster (depending on the number of detections) and more efficient than GA in a narrow range of situations or experiments. Sadly, its efficiency cannot be improved, as changing parameters do not change the result significantly. If I am wrong, please show it to me.

We respectfully disagree with the reviewer's conclusion. While earlier efforts have primarily focused on STORM and PALM data, G5M is specifically designed to make use of the unique characteristics of DNA-PAINT, addressing a clear gap in existing molecular mapping approaches. By using the well-defined binding kinetics and repeated sampling intrinsic to DNA-PAINT, G5M reliably outperforms alternative algorithms that have been or could be applied to identify molecule coordinates from DNA-PAINT localizations.

Across in silico benchmarks, in vitro DNA origami structures, and cellular datasets, G5M resolves molecules at shorter inter-molecular distances than competing methods while maintaining high true-positive rates and low false-positive rates. Importantly, G5M remains robust across user-defined parameters, ensuring consistent performance and making the algorithm accessible for non-expert users. Taken together, these results demonstrate that G5M is more accurate and more broadly applicable than currently available approaches.

Reviewer #3 (Remarks to the Author):

The manuscript introduces G5M, a probabilistic algorithm designed to extract molecular positions from DNA-PAINT single-molecule localisation microscopy data. The authors benchmark G5M against simulations of different molecular organisations, showing significant improvement over their previous gradient ascent (GA) methodology. Validation on simulated datasets, DNA origami structures, nuclear pore complexes, and CD20 receptor oligomers demonstrates that G5M reliably recovers known molecular architectures and reveals higher-order assemblies that are missed by the current GA approach.

By embedding G5M within the open-source Picasso platform, the authors make a sophisticated modelling framework broadly accessible to the super-resolution community. The anticipated impact of this work for the PAINT imaging community is significant: it provides a generalizable, statistically principled method for molecular mapping that enhances both the accuracy and interpretability of PAINT data. How widely this may be applied to SMLM data from methods other than DNA-PAINT remains to be seen, but the work is a strong addition to the DNA-PAINT analysis framework. I have only a few very technical reservations, which I have outlined below.

We thank the reviewer for their appreciation of our work.

Points of clarification:

The assumption that localisations are normally distributed may be reasonable for ideal datasets (e.g., minimal drift, uniform TIRF illumination). However, if linked localisations are used, the precision of the linked molecules may not follow a Gaussian distribution, since position averaging during linking will typically increase precision. How would this impact on the performance of the G5M algorithm? This should be briefly discussed to inform users of appropriate input data format.

We show that our localizations are in agreement with a normal distribution (**Fig. S1**), indicating the absence of drift and other artifacts. We agree that linked localizations, e.g., combining the localizations from a single binding event into one data point with higher resulting precision, will not follow a Gaussian distribution. The precisions of the linked localizations depend on the length of the binding event, and thus their distribution will not follow a Gaussian distribution with one unique standard deviation σ . Thus, G5M is not supposed to be applied to linked localizations. We now discuss the Gaussian nature of the distribution of localizations (but not of linked localizations) in the Discussion and the Supplementary Information (**Section 1 Single-Molecule Localizations are Gaussian distributed**).

It was not clear from the manuscript how localisation precision is defined. The provided code also does not prompt the user for this value, and it is unclear where it is extracted from. Is the localisation precision defined as the average fitting precision of individual localisations, or as the NeNA localisation precision? If the latter, the following reference should be cited: Endesfelder et al. *Histochem. Cell Biol.* 141, 629–638 (2014). This citation should also be added in the “DNA origami sample preparation and imaging” section of the Methods (page 15).

G5M uses the localization precision to choose the σ bounds ((0.8 – 1.5) LP). As referenced in the Concept and Algorithm section of the paper, G5M calculates localization precisions based on Mortensen et al. (*Nat Methods*, 2010). Using these localization precisions, G5M automatically calculates the average localization precision of a Gaussian component and subsequently the σ bounds. Only if the user decides against this default setting and actively chooses “custom σ bounds” for the complete dataset, appropriate bounds need to be identified. For this purpose, it is recommended to measure the average localization precision of the dataset using NeNA, which we already cited in the Supplementary Information. We now clarified the definition of localization precision in the **SI section 3.3.1 Local and global σ bounds**.

Molecular position precision in different conditions:

Please include a metric for the nearest-neighbour distance between the identified molecular positions and the ground truth positions in the simulated data (Figure 2). This would complement the reported relative detection numbers (Figures 2d and 2h) and provide a clearer measure of the G5M algorithm's performance. It would also be valuable to examine how the precision of molecular positioning depends on the average number of localisations collected per molecule.

We thank the reviewer for this comment. We evaluated precision of G5M by measuring the first nearest neighbor distances from the ground truth positions to the recovered positions for the dimer and 3x4 grid simulations and summarized the findings in **Fig. S7**.

Dependence on pre-clustering with DBSCAN:

As noted in the Supplementary Material (page 5), the performance of the G5M algorithm is highly dependent on the quality of the initial clustering. This represents a critical point where variability in data will necessitate user expertise in parameter selection. Page 8 encourages users to experiment with the DBSCAN parameters, but there is a lack of objective measurements to provide guidance on what constitutes a successful or unsuccessful clustering outcome.

The manuscript suggests that DBSCAN should be tuned to (i) remove background and retain "true" localisations, and (ii) divide the data into subsets containing fewer than 10 molecules. This advice assumes that users can reliably identify "true" localisations and know the approximate number of molecules contributing to each cluster; but this is precisely the quantities that the G5M algorithm is intended to infer. Consequently, this step introduces a high degree of subjectivity into the analysis.

Perhaps the authors could provide a simple tool to help users assess the DBSCAN clustering output. For example, they might include a function that quantifies the area of clusters produced by a user's defined parameter settings and reports the proportion of clusters that exceed a benchmark size. This could be the area corresponding to the tightest distribution of localisations that the G5M algorithm could fit ten molecules to, such as the combined area of ten circles with radii equal to half the Rayleigh limit (based on the given localisation precision), or another similarly justified threshold.

We thank the reviewer for the remark and the suggestion. We now systematically test the influence of DBSCAN parameters ϵ and *min. samples* on G5M outcome (**Fig. S8**), providing justification for the recommended ϵ values for 2D and 3D data respectively. Moreover, we showed that G5M outcome is largely robust to changes in *min. samples* on our data.

Based on the suggestion of the reviewer, we implemented a threshold for the DBSCAN cluster size. When crossed for more than 0.5% of DBSCAN clusters, the user is prompted to adjust parameters to reduce cluster size, allowing for faster and more accurate G5M analysis. Details are described in **Supplementary Information section 3.4**, including **Figure S18**.

A complementary diagnostic to evaluate background removal would also be valuable. For example, the authors could report the proportion of localisations classified as "noise" by DBSCAN for given parameters, or visualise how this fraction varies as parameters are adjusted.

We thank the reviewer for this comment. Following the suggestion, we modified Picasso to save the fraction of discarded localizations as a result of clustering. The .yaml metadata file stores this information. The values for our data are now reported in the **Methods**.

Additionally, an objective performance measure could be obtained by reconstructing images from (i) all localisations and (ii) only those retained as clustered, using pixel bins approximately half the localisation precision, and then comparing their Fourier Ring Correlation (FRC) resolutions. An improvement in FRC resolution for the clustered subset would indicate that background removal has produced a sharper and more spatially coherent signal. While FRC is not a direct measure of noise suppression, it would provide users with

a reproducible, quantitative means of assessing whether their parameter choices have removed background and enhanced data quality.

Following the suggestion of the reviewer, we tested FRC as a metric to measure background removal. We used the FRC function in the BIOP Fiji plugin (<https://imagej.net/plugins/fourier-ring-correlation>, based on Niewenhuizen et al (2013). *Measuring image resolution in optical nanoscopy*. *Nature Methods*, 10, 557.

We performed FRC on DNA origami prior to DBSCAN (Fig. R2a) and on the localizations retained by DBSCAN with different parameters (Fig. R2b-f). The dataset containing all the localizations (no DBSCAN) resulted in a FRC resolution (FIRE value) of 24.6 nm. Performing DBSCAN with an ϵ of three times the localization precision ($\epsilon = 3$ LP) already increased the FRC resolution to 9.6 nm. With $\epsilon = 2$ LP and $\epsilon = \sqrt{2}$ LP, DBSCAN removed localizations in the immediate vicinity of origami sites, resulting in a further improvement of resolution to 5.4 nm and 4.7 nm, respectively. However, even smaller ϵ search radii, like $\epsilon = 1$ LP or $\epsilon = 0.5$ LP, lead to the exclusion of a significant portion of localizations representing DNA origami sites, while at the same time showing the highest resolution values. So FRC measures the increased sharpness of the image, but cannot distinguish between noise removal or exclusion of localizations actually coming from a DNA-PAINT docking site. Therefore, FRC does not appear to be a robust metric to evaluate DBSCAN outcome within the G5M algorithm.

Figure R2: Fourier ring correlation for different DBSCAN parameters. To perform FRC, localizations were split into two random subsets and binned into images with pixel size equal to half the localization precision (LP = 1.6 nm, pixel size = 0.8 nm). The FIRE value (inverse spatial frequency at a correlation threshold of 1/7) was identified using the FRC function within the BIOP Fiji plugin (<https://imagej.net/plugins/fourier-ring-correlation>, based on Niewenhuizen et al (2013). “Measuring image resolution in optical nanoscopy”. Nature Methods, 10, 557). **a**, FRC of a raw dataset of 360 DNA origamis yields a FIRE value of 24.6 nm. **b-f**, The DBSCAN search radius ϵ was varied while min. samples = 4 was kept constant. FRC was performed on the localizations retained by DBSCAN. These localizations and the G5M-recovered sites are shown for an exemplary origami.

Additional validation metric:

It would also be useful to measure the influx rate within each DBSCAN cluster. Given known on- and off-rates for binding, this could provide an independent estimate of the number of molecules per cluster. Comparing this to the number of sites fitted by G5M would offer an indicator of underfitting (e.g., missing closely spaced sites), particularly when applied to experimental data.

We agree that qPAINT is a useful control to verify stoichiometries obtained by G5M. To demonstrate this, we performed qPAINT on the DNA origami dataset with median LP = 1.6 nm. In **Figure R3**, we plot the number of molecules recovered by G5M vs. qPAINT (following Jungmann et al. [[10.1038/nmeth.3804](https://doi.org/10.1038/nmeth.3804)]), obtaining an average of 12 sites per origami with both approaches.

However, qPAINT requires a careful calibration where individual binding sites are picked to extract the mean dark time for a single binding site (corresponding to a single biomolecule, e.g., a protein). If such a calibration measurement is provided, the user can perform qPAINT as a validation of G5M results. We now explain how the user can use qPAINT as a control metric for G5M in **Supplementary Information Section 3.2**.

Figure R3: qPAINT vs G5M comparison on DNA origami. Number of molecules recovered by G5M (vertical axis) and qPAINT (horizontal axis) in well-resolved DNA origami where single-site localization clouds were picked for calibration. The heatmap indicates the frequency of each combination of the numbers of the recovered molecules by both techniques on a logarithmic scale. The dashed line indicates the identity function.

Reviewer #3 (Remarks on code availability):

The code was relatively easy to install and implement in the Picasso environment. I particularly found parameterising the DBSCAN a little challenging and difficult to assess when I tried it on some data from my

lab. The lack of GUI for DBSCAN results and lack of definition of localisation precision made it feel like a substantial proportion of the process was buried "under the hood".

We have now clarified the metric used for localization precision in **Supplementary Information Section 3.3.1**. Regarding DBSCAN parameter assessment, Picasso provides a tool for testing clustering algorithms, including DBSCAN, on a small FOV with fast feedback. After loading data in Render, please choose Postprocess -> Clustering -> Test clusterer. This is now mentioned in **Supplementary Information Section 3.4**. We are further planning to add G5M to this tool once G5M is published.